# Karyotype variation, spontaneous genome rearrangements affecting chemical insensitivity, and expression level polymorphisms in the plant pathogen *Phytophthora infestans* revealed using its first chromosome-scale assembly

**Michael E. H. Matson[1], Qihua Liang[2], Stefano Lonardi[2], Howard S. Judelson[1]***

**1** Department of Microbiology and Plant Pathology, University of California, Riverside, California, United States of America, **2** Department of Computer Science and Engineering, University of California, Riverside, California, United States of America

* howard.judelson@ucr.edu

## Abstract

Natural isolates of the potato and tomato pathogen *Phytophthora infestans* exhibit substantial variation in virulence, chemical sensitivity, ploidy, and other traits. A chromosome-scale assembly was developed to expand genomic resources for this oomyceteous microbe, and used to explore the basis of variation. Using PacBio and Illumina data, a long-range linking library, and an optical map, an assembly was created and coalesced into 15 pseudochromosomes spanning 219 Mb using SNP-based genetic linkage data. *De novo* gene prediction combined with transcript evidence identified 19,981 protein-coding genes, plus about eight thousand tRNA genes. The chromosomes were comprised of a mosaic of gene-rich and gene-sparse regions plus very long centromeres. Genes exhibited a biased distribution across chromosomes, especially members of families encoding RXLR and CRN effectors which clustered on certain chromosomes. Strikingly, half of F1 progeny of diploid parents were polyploid or aneuploid. Substantial expression level polymorphisms between strains were identified, much of which could be attributed to differences in chromosome dosage, transposable element insertions, and adjacency to repetitive DNA. QTL analysis identified a locus on the right arm of chromosome 3 governing sensitivity to the crop protection chemical metalaxyl. Strains heterozygous for resistance often experienced megabase-sized deletions of that part of the chromosome when cultured on metalaxyl, increasing resistance due to loss of the sensitive allele. This study sheds light on diverse phenomena affecting variation in *P. infestans* and relatives, helps explain the prevalence of polyploidy in natural populations, and provides a new foundation for biologic and genetic investigations.

**Data Availability Statement:** Data are available at NCBI under Bioproject PRJNA868814.

**Funding:** This work was supported by grants to HSJ from the National Science Foundation and the USDA National Institute of Food and Agriculture and to SL from the National Science Foundation. The funders had no role in study design, data collection and analysis, decision to publish, or preparation of the manuscript.

**Competing interests:** The authors have declared that no competing interests exist.

## Author summary

Strains of the oomycete *Phytophthora infestans* from nature vary in traits including aggressiveness and chemical resistance, and range from diploid to tetraploid. What causes such variation in this important potato and tomato pathogen is largely unknown. To study this we generated the species' first chromosome-scale assembly and employed it to study karyotypic, transcriptomic, and phenotypic variation between strains. Intriguingly, we found that about half of progeny of diploid parents were polyploid or aneuploid. Many expression level polymorphisms between strains resulted from karyotype differences, transposon insertions, and the influence of repetitive DNA flanking genes. We also used SNPs to map loci determining sensitivity to a popular crop protection chemical, and found that a loss of heterozygosity affecting one locus often occurred which increased resistance. We also observed that the genome was organized into a montage of intermingled gene-dense and gene-sparse zones, with long pericentric regions.

## Introduction

A myriad of factors underlie phenotypic variation, including small genetic changes that modify gene expression or protein function and larger-scale phenomena affecting chromosome structure or ploidy. Understanding the genetic causes of such variation addresses fundamental topics such as evolutionary dynamics as well as issues of more practical importance. Genetic variation in a pathogenic microbe, for example, can influence its ability to infect hosts or be susceptible to chemical control [1]. Similarly, genetic variation in plants and animals affects their predisposition to disease and yield [2–4]. Technological advancements have facilitated a recent shift from locus-specific studies of variation to genome-wide assessments, preferably based on a high-quality genome sequence [5].

Most genome assemblies of oomycete plant pathogens, a group within the stramenopile clade of eukaryotes [6], have been highly fragmented due to a combination of novel genome architectures and genomes that are often large with high repetitive DNA content [7]. The most notorious of these filamentous microbes are in the genus *Phytophthora*, such as *P. infestans* which is responsible for the devastating late blight disease of potato and the broad host-range "dieback" pathogen *P. cinnamomi* [8]. While many oomycete genomes have modest sizes of about 35 Mb, others have swelled to about 250 Mb due largely to the expansion of sequences resembling transposable elements [9]. Much of this repetitive DNA is scattered throughout the genome, forming a mosaic of gene-dense clusters and gene-sparse zones [6]. Enriched in the repeat-rich regions are genes encoding effectors such as RXLR and CRN proteins, which pathogens use to suppress plant immune responses but which may also trigger host defenses if recognized by plants [10,11]. These genes have experienced accelerated evolution through family expansions and birth or death events, and can become epigenetically silenced, which helps to explain how some strains may evade host defenses [12].

Still, many aspects of phenotypic variation in oomycetes remain poorly understood. Within *P. infestans*, for example, isolates vary in their sporulation intensity, optimal temperature for spore germination, and aggressiveness, and sensitivity to chemicals used for crop protection [13–16]. Variation in ploidy has also been detected, but its origin and biological consequences remain obscure. *Phytophthora* is defined classically as a diploid genus in which species are either homothallic (self-fertile) or heterothallic with two mating types [17]. The sexual cycles involve the reduction of diploid somatic nuclei into haploids within male and female gametangia (antheridia and oogonia, respectively), which fuse to form one zygote per oospore.

However, several species have been shown to include polyploids [18,19]. For example, the majority of isolates of *P. infestans* from most geographic regions are triploid [20,21]. Studies of polyploidy initially used classical cytophotometric techniques, later shifting to methods such as flow cytometry and microsatellite fingerprinting [22–24]. Analyses based on single nucleotide polymorphisms (SNPs) became popular after the first genome assembly for *P. infestans* was released in 2009, based on Sanger sequencing [9,25].

The Sanger-based assembly of *P. infestans* has been a boon to the community as it spurred investigations of gene function, enabled strategies such as RNA-seq for analyzing gene expression, and provided a resource for studying population genetics [25,26]. It also confirmed reports that *P. infestans* has one of the largest oomycete genomes, with the assembly covering 228 Mb and containing approximately 74% repetitive DNA [9]. However, the assembly was highly fragmented and incomplete due to the large size and high repeat content of the genome, combined with limitations of the Sanger method.

The goal of the current study was to use modern sequencing technologies to build an improved genome assembly for *P. infestans*, and test its utility for addressing issues in oomycete biology. By integrating short and long-range reads with an optical map, a long-distance linking library, and genetic maps based on SNPs, we developed an assembly spanning 247 Mb with 15 chromosome-sized scaffolds. This was then used to characterize the organization of genes and repetitive DNA within chromosomes, study variation in chromosome structure including spontaneous deletions, and detect an exceptionally high frequency of polyploids and aneuploids within progeny of diploid parents. We also characterized expression level polymorphisms including those resulting from changes in ploidy or chromosome structure, and mapped a locus determining insensitivity to an important crop protection chemical, metalaxyl.

## Results

### Development of chromosome-scale assembly

We reported previously on a new assembly method that allowed us to produce a preliminary genome assembly for strain 1306 *of P. infestans* based on Pacific Biosciences reads, Illumina reads, a long-range linking library using the Dovetail Chicago method, and an optical map based on the Bionano platform [27]. That assembly contained 250.4 Mb distributed over 1,329 contigs with an N50 of 0.73 Mb. We refined that assembly using an additional round of analysis using Dovetail software, resulting in a 246.5 Mb assembly (hereafter called Stitch7) encompassing 1,034 contigs (after removing mitochondrial DNA) with an N50 of 0.97 Mb. To check the completeness of Stitch7, we assembled RNA transcripts from RNA-seq data using the Trinity program. Of the 25,881 resulting transcripts, 99.6% mapped to the assembly using BWA-mem [28], indicating a high degree of genome completeness. Moreover, using the stramenopile gene set with BUSCO [29] we identified 100% complete genes with 7% duplications. The size and continuity of this assembly compares favorably with those of other strains of *P. infestans* (S1 Table).

The Stitch7 assembly was next used to evaluate whether isolate 1306 was diploid. This involved using our Illumina data to score the frequency of alternate alleles of single nucleotide polymorphisms (SNPs). Within the assembly, 246,251 heterozygous SNP loci were identified. Reads representing the alternate variants occurred at an average ratio of 1:1, consistent with diploidy (Fig 1).

We then consolidated the 1,034 scaffolds from Stitch7 into pseudochromosomes (referred to hereafter as chromosomes) by genetic analysis using the F1 progeny of two crosses, 1306 × 618 and 6629 × 550. Like 1306, the other three parents were judged as being diploid by

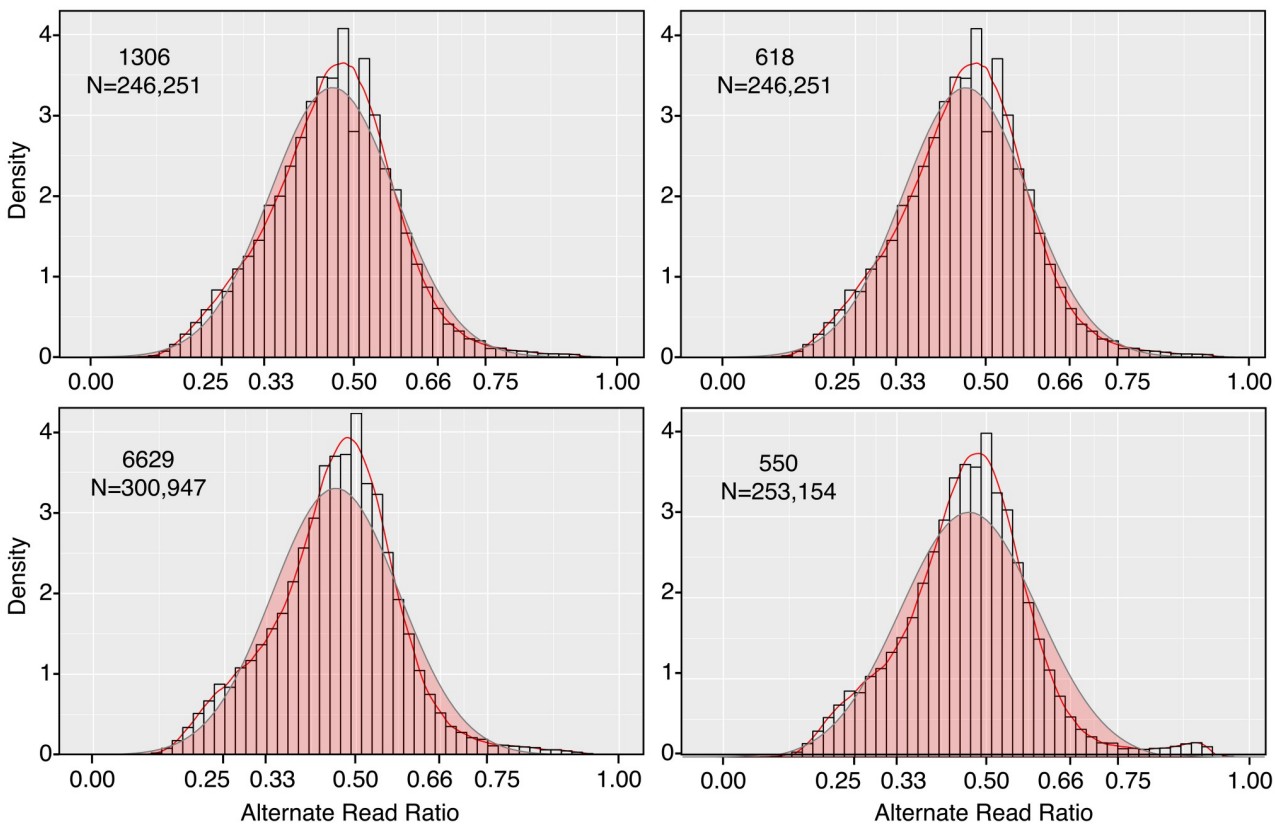

**Fig 1. Alternate read ratios in parents of crosses.** Heterozygous SNPs were identified in strains 1306, 618, 6629, and 550 by mapping Illumina reads to the Stitch7 assembly. Sequence variants (alternate reads) were identified and their ratio distributions modeled as described in Methods, with the *y*-axis referring to kernel density which is plotted as a red line. Also in the plot is a normalized distribution of the data (red fill, grey line), and the relative frequency of variants per bin (histograms). The number of SNPs scored for the strains are indicated in the panels.

SNP ratio analysis, since their alternate alleles occurred at an average ratio of 1:1 based on mapping Illumina reads to the Stitch7 assembly (Fig 1). For linkage analysis, we identified pseudo-testcross markers from each parent using small indel variants and SNPs. After manual curation and subsampling to every 5 kb, we selected 2,039, 2,108, 874, and 1,202 markers from strains 1306, 618, 6629, and 550, respectively. Genetic maps were then constructed for each parent using JoinMap [30] with data from 82 sequenced F1 strains from the 1306 × 618 cross and 32 from the 6629 × 550 cross. This resulted in 15, 15, 16, and 17 linkage groups spanning 1,870, 1,289, 1,608, and 1,658 cM in strains 1306, 618, 6629, and 550, respectively, with an average of one marker every 35.4 kb. These diploid sizes compare to a total of 827 cM spread over 10 major and 7 minor linkage groups reported for polyploid Dutch strains in a mapping study using AFLP markers [31]. Cytological studies of *P. infestans* have reported varying numbers of chromosomes in *P. infestans*, with up to 12 described in one study [32].

Since DNA from strain 1306 was used to construct the assembly, its linkage groups were used as the primary guide in building chromosomes. This resolved 15 chromosome-sized scaffolds ranging in size from 10.1 to 22.9 Mb, and 47 to 498 cM (Fig 2). Linkage and recombination data for six chromosomes are illustrated in Fig 3, and for all chromosomes in S1 Fig. These encompassed 89% of the DNA in the Stitch7 assembly including 417 of its 1,034 contigs. Eleven chromosomes contained regions that lacked sequence coverage but were predicted by the optical map to cover between 110 and 660 kb (″g″ labels in Fig 2A). These may represent

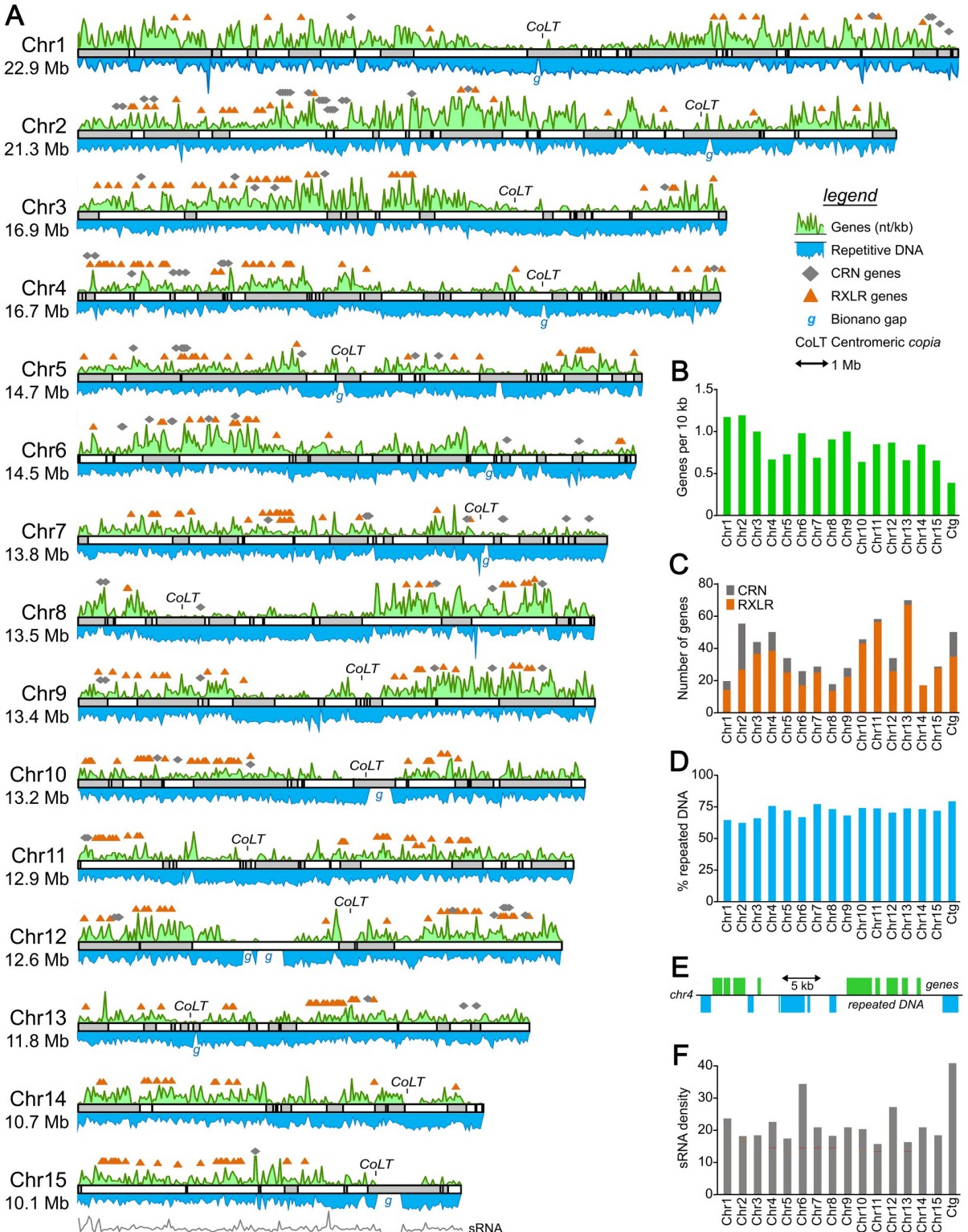

**Fig 2. Structure of *P. infestans* chromosomes. A**, Maps of the chromosomes, with contigs from the Stitch7 assembly shown as alternating white and grey blocks. Each chromosome (Chr prefix) is marked along with its predicted size in Mb. The green and blue curves above and below each chromosome map represent the relative density of genes and repeats, respectively, as nt per kb. Positions of effector genes are shown above each chromosome, with RXLRs and CRNs represented by orange triangles and grey diamonds, respectively. Clusters of the *copia*-like element associated with centromeres are marked as *CoLT*, and gaps predicted by the Bionano optical map are marked as *g*. The grey

line under Chr15 represents the relative density of small RNAs, as nt per kb. **B**, gene densities per chromosome (Chr1 to Chr15) or on contigs not placed on chromosomes (ctg). **C**, number of CRN and RXLR genes per chromosome or unplaced contigs. **D**. Percent repeated DNA per chromosome or on unplaced contigs. **E**, typical arrangement of genes and repeated DNA, showing a representative portion of Chr4. **F**, relative density of small RNAs on chromosomes or contigs.

missing portions of centromeres, which are notoriously difficult to assemble. Consistent with this premise, detected near most of these gaps were clusters of the CoLT *copia*-like retroelement that had been shown to reside near *Phytophthora sojae* centromeres [33]. No CoLT cluster was detected on Chr6, however. This raises the possibility that Chr6 should be fused with another chromosome. Telomeric sequences were detected near the ends of Chr2, 7, and 8. Most contigs that could not be placed into chromosomes were small, averaging 42.6 kb, and enriched for repetitive sequences.

## Chromosome structure varies between the parents

Structural differences between the parental chromosomes were inferred from the data. In particular, some exhibited "forked" recombination plots consistent with rearrangements such as inversions. One example shown in Fig 3 is Chr5, where the 6629 parent appears rearranged compared to parents 1306, 618, and 550. Forked plots were also seen for Chr3 and Chr11.

While markers from all four parents were typically distributed among all chromosomes, there were exceptions which may signal other rearrangements, deletions, or losses of heterozygosity. For example, few loci from parents 6629 or 618 could be placed on Chr6 or Chr8, respectively. Cases where markers from 6629 or 550 could not enter a single linkage group (*e.g.* Chr1) might also indicate structural variation. An alternative explanation may be that the limited number of 6629 × 550 progeny reduced the number of markers that could be placed confidently into a linkage group.

Higher rates of recombination occurred near both termini on at least nine chromosomes; examples in Fig 3 include Chr5 and Chr11. This suggests that these have a metacentric configuration since recombination is suppressed at centromeres in most species [34]. In other cases, recombination was reduced near only one end, suggesting a more telocentric arrangement (*e.g.* Chr3 and Chr7). The regions of reduced recombination were usually gene-sparse, as noted in the next section.

## Gene distribution between chromosomes is uneven

Gene models were developed primarily using Funannotate [35], which integrated *de novo* gene prediction with transcript evidence from hyphae, sporangia, zoospore, and germinated cyst life stages, and gene models identified previously from *Phytophthora parasitica*, *Phytophthora sojae*, *Pythium ultimum*, and strain T30-4 of *P. infestans*. This predicted 19,891 protein-coding genes, which averaged 1,350 bp in size with 2.76 exons per gene. The GC content of their exons, introns plus 5' and 3' untranslated regions, and intergenic regions were 53.8%, 49.5%, and 50.3%, respectively. To check the completeness of the Funannotate results, its predicted proteins were analyzed using BUSCO [29] with the stramenopile gene set. This revealed 98% completeness with 0% fragmentation. The missing genes could be identified by performing a lift-over of gene models predicted previously from T30-4 [9] that lacked matches in the output of Funannotate. Overall, matches in the PANTHER, INTERPRO, or PFAM databases were identified for 69% of the proteins. RXLR and CRN effectors were encoded by 549 and 128 genes, respectively.

Previous studies have shown that oomycete genomes are a mosaic of gene-dense and gene-sparse regions, the latter being rich in repetitive DNA [6]. About 93% of genes in strain 1306

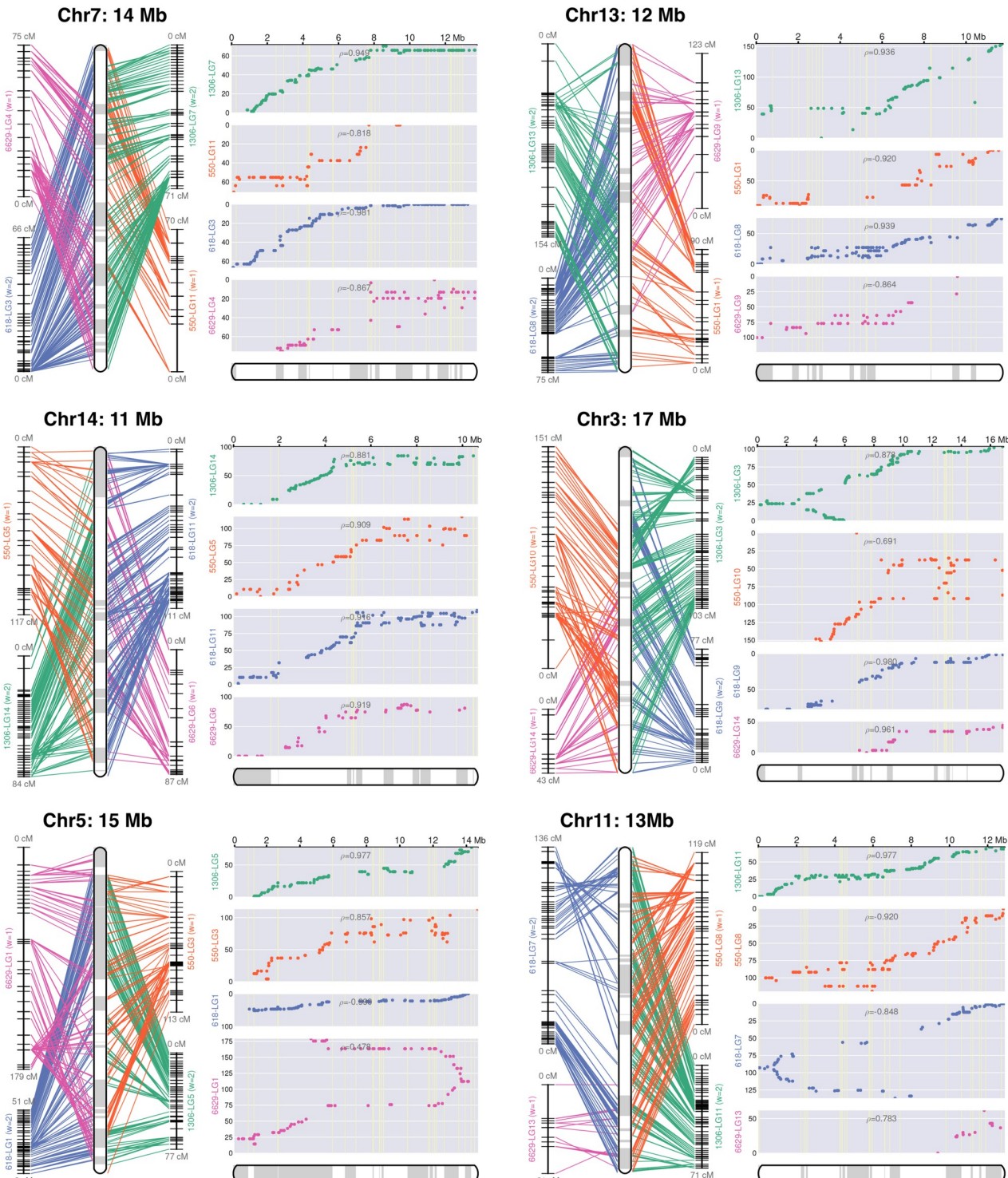

**Fig 3. Representative chromosome assemblies.** Illustrated are genetic and physical maps for six of the chromosomes; all 15 are presented in S1 Fig. The left panel for each chromosome shows line graphs representing linkage groups (LG) developed for the four parents, with horizontal marks representing loci used for genetic linkage analysis. The rounded rectangles at the center of the panel represent the consensus map for the chromosome, with contigs joined by linkage analysis represented by alternating white and grey sections. The right panels (grey background) show plots of genetic versus physical distance for each chromosome in each parent. The *x*-axis represents the physical location along the chromosome and the *y*-axis represents the distance in centimorgans.

reside in gene clusters, each containing an average of five genes. In this context, a cluster is defined as genes spaced within 2.5 kb of each other, which may or may not have related functions or act in the same pathway.

The unevenness of gene distribution becomes more evident when examined at the higher-level scale of whole chromosomes (green graph in Fig 2A). Genes were distributed nonuniformly between the chromosomes, ranging from a high of 1.15 per 10 kb for Chr1 to a low of 0.62 for Chr10 (Fig 2B). Variation was also observed along each chromosome. For example, gene density was much higher at the left and right sides of Chr1 compared to its center, which we presume contains a centromere based on its low gene density and the presence of the CoLT sequence. There was a positive association between gene density and recombination rate.

The distribution of genes encoding RXLR and CRN effectors was more biased than total genes (Fig 2C). For example, while RXLR genes represented only 0.6% of total genes on Chr1, they accounted for 5.9% of genes on Chr13. The prevalence of CRN genes also varied, ranging from 1.2% of genes on Chr2 to none on Chr4. Nearly all grouped into clusters on the chromosomes (Fig 2A). Most clusters were diffuse and spanned a few megabases, suggesting their evolution through unequal crossing-over, and rearrangements which might include transposon-mediated expansion. This model is congruent with an analysis of the distribution of effector subgroups defined previously by TribeMCL and domain analysis [9]. For example, all members of RXLR family 5 and 9 clustered on Chr5 and Chr4, respectively. There was also evidence for movement between chromosomes, for example with RXLR family 1 in which most members resided on Chr10 and Chr13. Similar patterns were observed for CRNs. For example, while all nine genes encoding members of the DC family resided on Chr4, and all four members of the DWL_DN5 group resided on Chr9, the DWL_DXZ group was split between Chr3 and Chr6.

## Transfer RNA gene numbers vary between *P. infestans* and other oomycetes

Using tRNAscanSE [36], 7,519 genes encoding tRNAs were predicted on the 15 *P. infestans* chromosomes, with an additional 3,537 on the 617 contigs not associated with chromosomes. Approximately 64% were part of tandem arrays of the same gene, with 36% belonging to arrays of dimeric or trimeric repeat units. The latter classes either encoded tRNAs for distinct amino acids (*e.g.* tRNA$^{TrpCCA}$ and tRNA$^{SerGCT}$) or the same amino acid with different anticodons (e.g. tRNA$^{GlnCTG}$ and tRNA$^{GlnTTG}$). Dimeric and trimeric arrays of tRNAs have also been reported in some other species, being infrequent in fungi but common in some plants [37,38]. About two-thirds of the tRNA families in *P. infestans* resided on a single chromosome (S2A Fig). For example, all 271 copies of tRNA$^{CysGCA}$ resided on Chr6, comprising a tandem array with a 513-nt repeat unit. Similarly, nearly all 85 copies of tRNA$^{ValAAC}$ and tRNA$^{SerAGA}$ were on Chr11 in the form of a dimeric 850-nt repeat. Most tRNA genes that were dispersed on multiple chromosomes also existed as tandem repeats.

The number of tRNA genes in *P. infestans* appeared high based on a prior study of the *P. ramorum* and *P. sojae* genomes, which were reported to contain 155 and 238 tRNA genes, respectively [39]. Most other species have also been described as containing fewer tRNA genes than indicated by our results with *P. infestans*, such as *Arabidopsis thaliana*, humans, and *Saccharomyces cerevisiae* with 631, 513, and 275 tRNA genes, respectively [40,41]. Consequently, we confirmed independently the number of tRNA genes in *P. infestans* using Illumina read depth analysis, finding concordance between those results and predictions from the assembly (S2B Fig). In retrospect, the prior estimates for *P. ramorum* and *P. sojae* were undercounts since they were based on a Sanger assembly, where tandem repeats had likely overcollapsed.

Indeed, using tRNAscanSE we identified 1,506 tRNA genes in a newer assembly of *P. ramorum* based on PacBio technology [42]. Assembly challenges may still persist when using long reads, however. This is suggested by the fact that 3,705 predicted tRNAs (34% of the total) resided on the unplaced contigs in our 1306 assembly. We suspect that many of these come from divergent haplotypes that did not enter the main assembly graphs.

The *P. infestans* tRNA genes matched only 51 codons, which suggests that wobble base-pairing is common. This may also explain why the number of tRNA genes and their cognate codons in protein-coding sequences were only modestly correlated (Pearson's $R$ = 0.41; S2C Fig). Also predicted were 20 potential suppressor tRNA genes and one encoding a putative selenocysteine tRNA.

To compare the number of tRNA genes in different oomycetes, we applied tRNAScanSE to genomes deposited in NCBI. To minimize artifacts due to overcollapsed assemblies, this was limited to genomes constructed with long reads from the Pacific Biosciences or Nanopore long-read platforms. The number of tRNA genes predicted ranged from a minimum of 380 for *Plasmopara halstedii* to a maximum of 8,606 for *Sclerospora graminicola* (S3A Fig). We observed a strong positive correlation (Pearson's $R$ = 0.72) between genome size and the number of tRNA genes, although this relationship weakened ($R$ = 0.22) when only genomes smaller than 100 Mb were considered. No consistent trend was observed between the downy mildew, *Phytophthora*, and *Pythium* (including *Globisporangium*) groups.

To help understand the differences in tRNA gene abundance between the species, we calculated the fraction corresponding to each amino acid (S3B Fig). The results indicated that some families had expanded preferentially in some species. For example, the proportions of genes encoding tRNA$^{Ser}$ and tRNA$^{Pro}$ were more than two-fold higher in *P. infestans* than most other species, due mostly to the enlargement of families with UGU and UGG anticodons, respectively. Similarly, the relative abundance of genes for tRNA$^{Met}$ and tRNA$^{Lys}$ was higher in *Aphanomyces cochlioides* and *P. cinnamomi*, respectively, than most other species. The biological impact of such expansions are unclear since the concentration of each tRNA may not be proportional to the copy number of its corresponding genes. Consistent with this premise, the most highly-expanded tRNA$^{Pro}$ and tRNA$^{Ser}$ families in *P. infestans* bind codons that are infrequently used based on codon usage tables.

These interspecific comparisons must be treated with caution due to the possibility of artifacts. While the data were limited to genomes incorporating long reads, the assembly programs as well as the sequencing technologies may influence the number of detected tRNA genes. For example, within *P. infestans* the number of predicted tRNA genes varied by about one-quarter between our assembly of strain 1306, a Nanopore-based assembly of strain KR2, and a PacBio-based assembly of strain RC1-10. Similarly, while the 58.6 Mb assembly of *Peronospora effusa* shown in S3 Fig yielded 7,391 tRNA genes (NCBI accession GCA_021491655; [43]), only 270 were identified in a 32.1 Mb assembly of the same species developed with PacBio reads (accession GCA013122855).

We also examined sequence diversity within each tRNA array within *P. infestans*, since this might shed light on how repeats had expanded within its genome. Three general patterns were observed. When the array had monomeric repeat units and occurred on a single chromosome (*e.g.* tRNA$^{LeuAAG}$, S4A Fig), each repeat was nearly identical. The few sequence variants within such arrays tended to occur in groups, often near one edge of the array. In contrast, a pattern of much higher diversity was detected for tRNA genes residing in arrays of dimeric and trimeric repeats, such as the tRNA$^{ThrCGT}$-tRNA$^{TrpCCA,}$-tRNA$^{SerGCT}$ array shown in S4B Fig. A third pattern was observed when tRNA genes with the same anticodon resided as monomeric arrays on separate chromosomes. In such cases, the sequences on the different chromosomes

were highly diverged with only modest differences within each chromosome (*e.g.* tRNA$^{MetCAT}$ on Chr8 and Chr12, S4C Fig).

## Other repetitive sequences comprise a supermajority of the genome

About 70.7% of the assembly was classified as repeats by RepeatModeler and RepeatMasker [44]. The true amount would be 72% if the gaps identified by the optical map were counted as repeats. Unlike the genic distribution, only minor variation in repeat content existed between chromosomes, ranging from a low of 62% for Chr2 to a high of 77% for Chr7 (Fig 2D). Consistent with prior analyses of strain T30-4 [9], the majority of the repetitive DNA resembled long terminal repeat (LTR) retrotransposons. These represented 46% of the assembly, with 48,378 copies identified (S5 Fig). The next largest annotated class were DNA transposons, with 28,031 copies representing 8.5% of the assembly.

The organization of repetitive DNA was variable across each chromosome (blue graph in Fig 2A). Genes and repetitive DNA were usually interspersed. This is difficult to discern from the density graphs in Fig 2A as their 50-kb window blurs the relationship between genes and surrounding repetitive DNA. However, shown in Fig 2E is a representative portrayal of the situation in most of the genome. In this 34-kb region from Chr4, two clusters of three and five genes, respectively, and a singleton, are separated by sequences having similarity to retroelements and DNA transposons.

A large repeat-rich, gene-sparse, recombination-suppressed domain that is probably the centromere was found on nearly all chromosomes. While centromeres in some species are transcriptionally active, hardly any RNA-seq reads mapped to these regions in *P. infestans* (S6 Fig.). The largest repeat-rich, gene-sparse, and transcript-scant features were on Chr1 and Chr8, each spanning about 4.5 Mb. While several of the putative pericentric features in *P. infestans* were more modest in size, overall they are nearly ten-fold larger than those in *P. sojae* and the downy mildew *Pe. effusa* [33,43]. Our examination of the pericentric region in Chr1 indicated that it contained about 95% repetitive DNA, not counting a gap in the assembly. About 86% and 9% resembled LTR/*gypsy* and LTR/*copia* elements, respectively, with much of the latter comprising a contiguous 175 kb domain.

About 1% of the1306 genome was represented by simple sequence repeats (SSRs) containing repeat units up to 11 nt in size. Their distribution was explored since SSRs are used frequently in studies of the population genetics and epidemiology of *P. infestans* [23]. Of the 19 SSR primer sets described in the literature, all but marker SSR8 matched targets in 1306 (S7 Fig.). While SSR markers used for population studies would ideally be distributed throughout the genome, none mapped to Chr1, Chr7, Chr 9, Chr11, or Chr15 while several were tightly clustered. Also shown in S7 Fig are sites corresponding to the multi-locus RFLP marker RG57, which had also been used for population genetics studies [45].

## Small RNAs map mostly to a subset of repeat-rich regions

Considering that small RNAs (sRNAs) often match transposable element-like sequences, which represent more than half of the *P. infestans* genome, we generated a library of sRNAs from strain 1306 and investigated their distribution across the chromosomes. Their hits were distributed unevenly through the genome, as shown in the density graph for Chr15 in Fig 2A and the per-chromosome summary in Fig 2F. As discovered in prior analyses [46, 47], the sRNAs fell into categories of roughly 21 and 25-nt with the latter being more common (S8B and S8C Fig). Detailed density graphs for all chromosomes are presented in S8 Fig panel A, showing total reads (unfiltered by mapping quality score in order to retain reads matching repeated sequences) and uniquely mapping sRNAs separated by size class.

Small RNAs associated with a range of features in the genome. Very few mapped to the putative centromeric regions, instead aligning to bins containing both genes and repetitive DNA. Only 11.8% mapped within genes themselves, however. A common pattern is shown in S8D Fig, representing the 50-kb to 110-kb interval from Chr14. This interval bears two small gene clusters flanked by retroelement and *Mutator* transposon-like sequences, with most RNAs matching the nongenic regions.

## Karyotypes resulting from sexual recombination vary

Strikingly, about half of progeny from both crosses were polyploid or aneuploid. This conclusion was based on karyotypes assessed by integrating allele ratio data (*i.e.* alternate read ratios) inferred from Illumina sequences with read depth data across the 15 chromosomes. The results indicated that the progeny included diploids, triploids, tetraploids, and aneuploids deficient (*e.g.* 2N-, 3N-) or having excesses (*e.g.* 2N+, 3N+) at one or more chromosomes (Fig 4A). Aneuploidy occurred more often for some chromosomes than others (Fig 4B). For example, Chr13 was most prone to aneuploidy while this was never observed for Chr12.

Representative data supporting these conclusions for the 6629 × 550 cross ("cross 20") are shown in Fig 5. For the 6629 and 550 parents, read depths were uniform across the 15 chromosomes (Fig 5A). Taking into account our observation that Illumina reads representing variants at heterozygous loci occurred at an average ratio of 1:1 (Fig 1), we infer that 6629 and 550 are diploids with balanced sets of chromosomes. An F1 strain that is also judged to be a normal diploid, 20.16, is illustrated in Fig 5B. Like its parents, this has a balanced set of chromosomes since read depths are even across the genome (Fig 5B). Moreover, diploidy is indicated by the occurrence of alleles from 6629 and 550 at an average ratio of 1:1. This is shown in the bar graphs in the lower part of Fig 5B, which display the relative densities of alleles (*i.e.* Illumina read variants) for the entire genome (left) and a representative chromosome (Chr6, right).

Examples of non-diploid progeny are in the remaining panels of Fig 5. For example, Fig 5C shows F1 strain 20.23, an aneuploid diploid (2N1-). Based on the read depth scatterplot, the dosages of Chr7 and Chr8 are approximately half of normal. That these are 1N is further evidenced by the bar graphs, which indicate that for Chr8 this strain contains alleles inherited from 6629 (1:0 ratio) but not 550 (0:1 ratio). In contrast, Chr3 has a normal dosage of chromosomes based on read depth and a 1:1 ratio of 6629 and 550 alleles. Thus, 20.23 is a 2N aneuploid.

Two aneuploids that are mostly 2N but bear three copies of some chromosomes are displayed in Fig 5D and 5E. With F1 strain 20.02 in Fig 5D, for example, the read depth scatterplot suggests the presence of extra copies of Chr3, Chr4, Chr9, Chr11, Chr13, and Chr14, and possibly a deficiency at Chr12. This is supported by the allele ratio graphs; while Illumina reads distinguishing Chr2 alleles from 6629 and 550 exhibit a 1:1 ratio consistent with diploidy, Chr9 appears to be trisomic based on a 2:1 ratio. A related pattern is shown in Fig 5E for F1 20.49, which is trisomic for Chr10 but diploid for Chr2. In analyses of the aneuploids such as these, we noted that the chromosomes present in extra copies did not always show the same quantitative increase in copy number (*e.g.* the number of mapped reads). For example, while both Chr9 and 11 in F1 20.2 have more reads than average, the read density for Chr9 is slightly higher (Fig 5D). Such patterns possibly indicate instability of the nuclear condition during culturing of the strains.

A mostly 3N strain is shown in Panel 5F. The read depth scatterplot for this strain, 20.03, shows only minor chromosome-to-chromosome variation. Yet a 2:1 ratio of alleles from the 6629 and 500 parents was seen across the entire genome, which is consistent with triploidy.

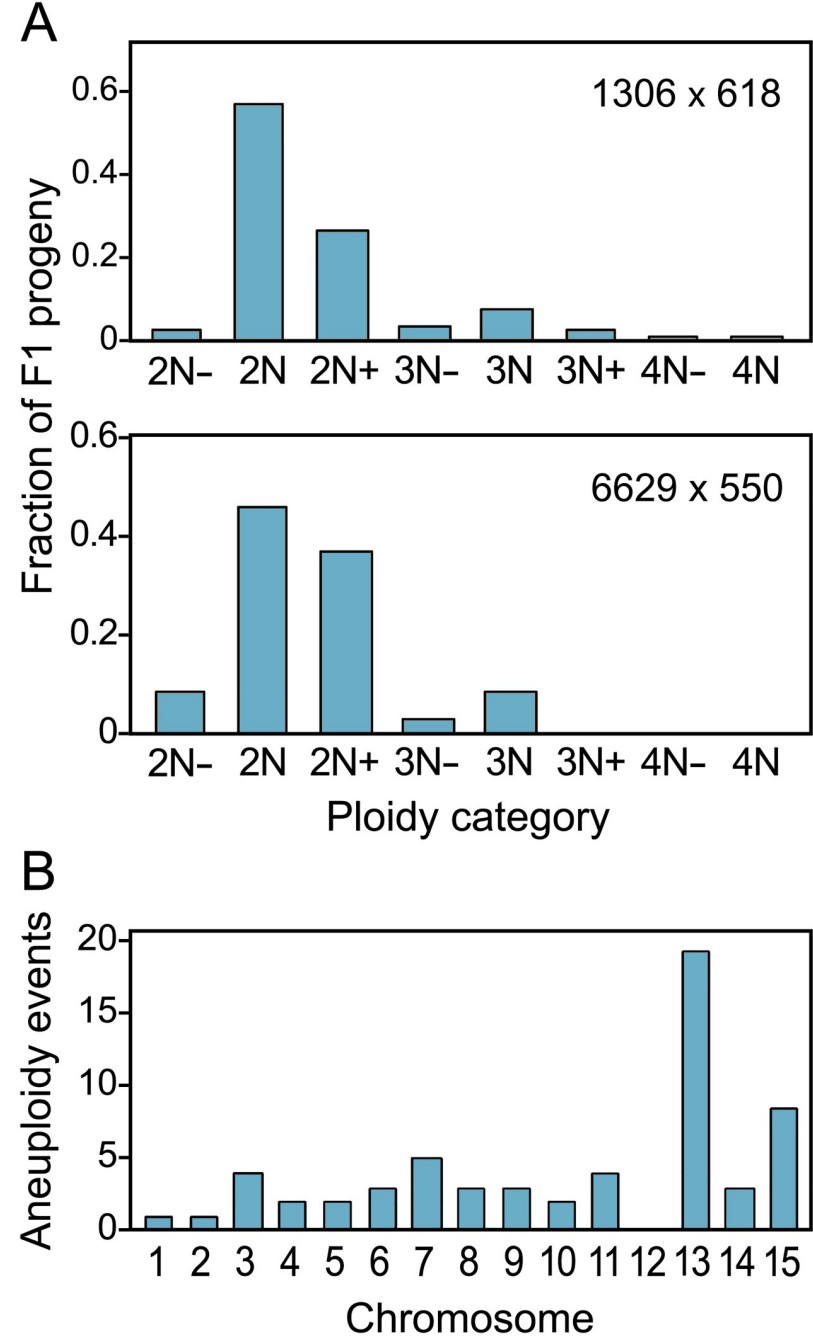

**Fig 4. Frequency of euploid and aneuploid progeny. A**, indicated for the two crosses are the relative ratios of diploid (2N) progeny, mostly diploid progeny that have chromosomal deficiencies (2N-) or excesses (2N+), triploids (3N) and their aneuploid variants (3N-, 3N+), and tetraploids (4N) and their aneuploid variants (4N-). Aneuploids are classified by whether they are closest to 2N, 3N, or 4N. **B**, Number of aneuploid events observed per chromosome.

To confirm our method for assessing ploidy, we used qPCR to score the copy number of chromosomes in two 2N+ aneuploids, F1 strains 20.02 (Chr9 trisomy) and 20.49 (Chr10 trisomy). This employed primers specific for loci on Chr5 (as a 2N control), 9, and 10. As shown in Fig 5G, the results were consistent with the conclusions drawn from the Illumina read

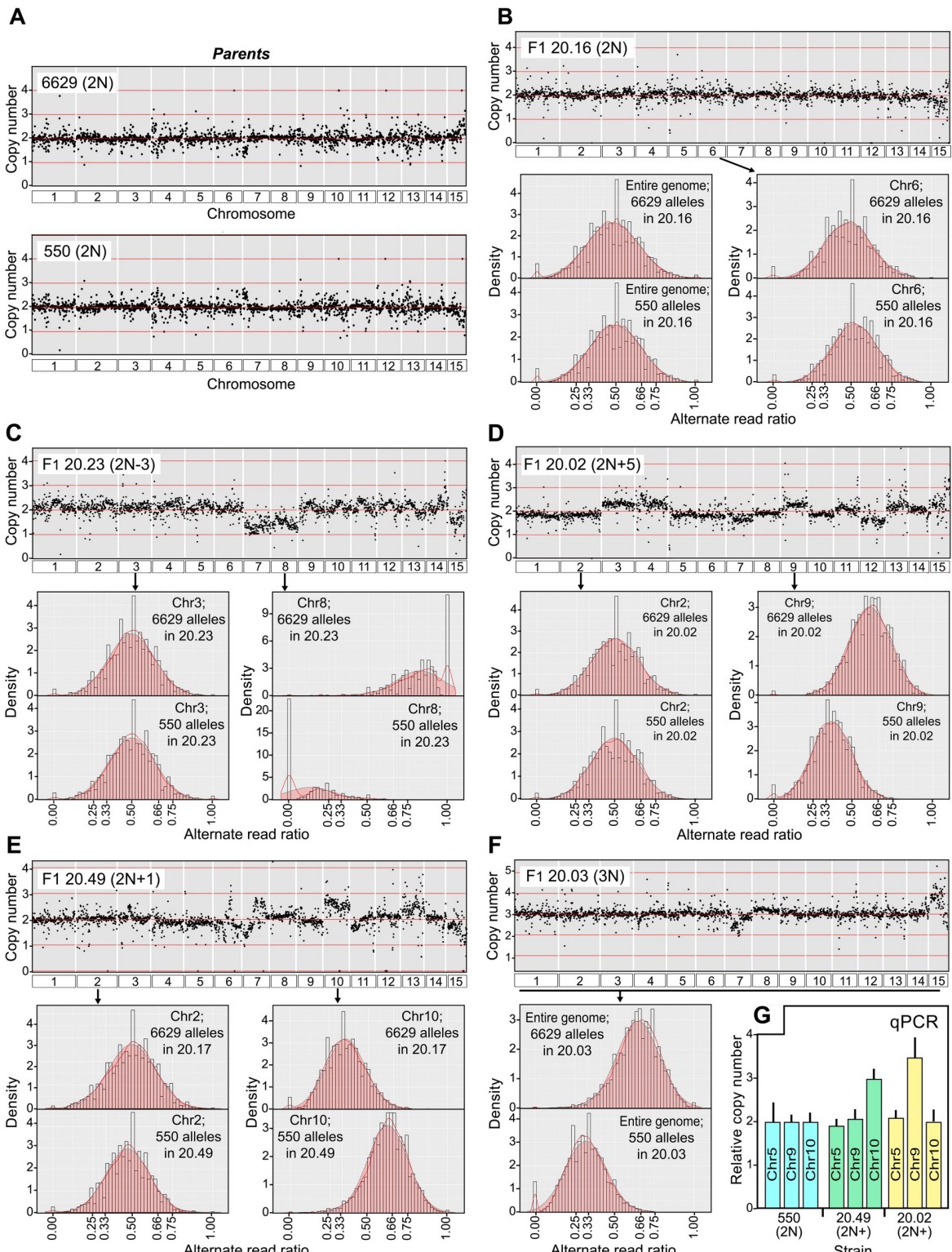

**Fig 5. Chromosomal content of representative strains. A**, Normalized Illumina read depth across all chromosomes in parents 6629 and 550. **B**, A representative diploid F1 (20.16) from the 6629×550 cross. Plotted in the scatterplot at the top is the normalized read depth. Presented below are bar graphs showing the alternate read ratios (i.e. inferred allele ratios) of SNPs originating from 6629 and 550 across the whole genome (left) or on a representative chromosome, Chr6 (right). **C**, Data for an aneuploid F1, 20.23, which bears Chr7 and 8 deficiencies. Read ratio graphs are shown for a typical 2N chromosome (Chr3) and one with a deficiency (Chr8). **D**, Similar data for F1

20.02, which appears to be mostly diploid with excesses (Chr3, Chr4, Chr7, Chr9) or deficiencies (Chr12). Read ratio graphs are shown for a representative 2N chromosome (Chr2) and one with three copies (Chr9). **E**, Similar data for $F_1$ 20.49, which appears to be diploid but with trisomy for chromosome 10. **F**, Similar data for $F_1$ 20.03, which appears to be a balanced triploid based on a 2:1 alternate read ratio across the genome. **G**, Validation of ploidy differences by qPCR. Primers targeting loci on Chr5, Chr9, and Chr10 were used to determine their copy numbers relative to strain 550 in progeny 20.49 and 20.02, which as indicated in panels E and D contain extra copies of Chr9 and 10, respectively.

depth and allele ratio data. For example, 20.02 and 20.49 were confirmed to an extra copy of Chr9 and Chr10 compared to 2N parent 550, respectively, while the Chr5 target was equally abundant (2N) in all strains.

## Changes in karyotype have modest effects on growth

Whether polyploidy or aneuploidy had major effects on *P. infestans* biology was assessed by scoring radial growth and sporulation rates on rye-sucrose agar, and lesion development on potato tubers. This involved phenotyping 76 of the F1 progeny from the 1306 × 618 cross (Fig 6). ANOVA was used to check for significant differences between diploids, triploids, tetraploids, and aneuploids, with the latter partitioned into 2N+,3N- or 3N+,4N- categories in the figure. On agar media the diploids grew on average 15% faster than strains in the other ploidy categories ($P = 0.0008$). No statistically significant variation based on the nuclear state was observed for sporulation ($P = 0.23$) and lesion development on tubers ($P = 0.11$). The sensitivity of our analysis was however constrained by the small number of progeny containing nuclear contents above 3N. It remains to be determined whether other stages in the disease cycle are affected by ploidy, such as appressorium development which occurs on leaves and not tubers.

Variation within each ploidy category was larger than that between categories. For example, even within the diploid group three-fold variation in growth rate and fifty-fold differences in sporulation intensity were observed. While some of variation between aneuploids might be

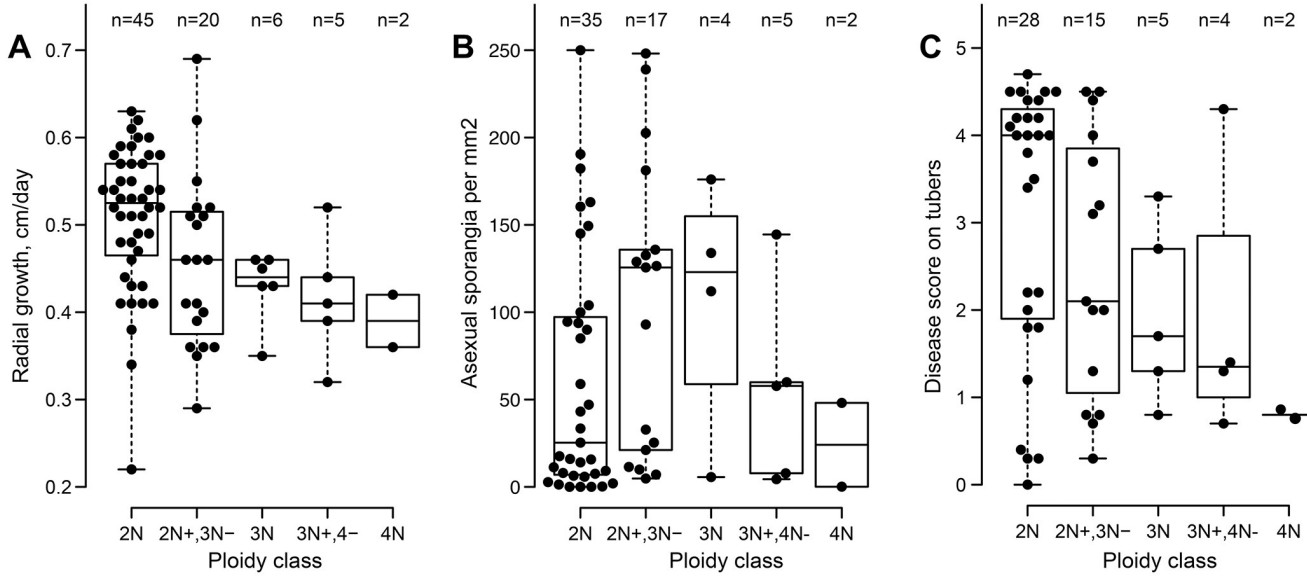

**Fig 6. Phenotypes in progeny.** F1 offspring of the 1306 × 618 parents were scored for **A**, radial growth rates on rye-sucrose agar; **B**, the intensity of asexual sporulation; and **C**, pathogenicity on potato tuber slices. Progeny are classified as diploids (2N), aneuploids between the diploid and triploid states (2N+, 3N-), triploids (3N), aneuploids intermediate to triploidy and tetraploidy (3N+, 4N-), and tetraploids (4N).

attributable to the specific unbalanced chromosome, the diversity witnessed in all groups suggests the involvement of other factors, such as strain-specific gene content or expression level polymorphisms (ELPs).

## ELPs between parents and progeny are common

RNA-seq analysis revealed expression level differences between 1306, 618, and their F1 progeny (Fig 7A). This involved examining RNA from mycelia grown in rye-sucrose broth, using two independent cultures per strain. The variation between the 32 F1 strains tested was often greater than that between the parents.

Polyploidy could explain some of this variation. Exhibiting significant differences between diploids and triploids were 215 genes, based on a Bonferroni-Hochberg false discovery rate (FDR) threshold of 0.01. As shown in the volcano plot in Fig 7B, it was more common for genes to have higher than lower mRNA levels in polyploids. Annotated functions of the genes with ploidy-associated ELPs are listed in S3 Table, with over-represented GO, INTERPRO, and KEGG terms shown in S4 Table. As indicated in the pie chart in Fig 7C, a substantial fraction (18%) encoded proteins with roles in mitosis. These included orthologs of the NimA protein kinase and MOB1 kinase regulators, which are associated with mitotic regulation, the MAD2A mitotic spindle checkpoint protein, the PSF2 replication complex protein, a mitotic kinesin, and others. That strains with increased nuclear content would require higher levels of such proteins is not surprising.

Aneuploidy also influenced gene expression. This is shown in Fig 7D which displays mRNA levels as FPKM (fragments per kilobase of exon per million mapped reads) of genes on Chr4, Chr13, and Chr14 in parent 1306 compared to representative diploid and aneuploid progeny of the 1306 × 618 cross. Average FPKM values in 2N F1 strains M76 and M65 are similar to 1306. However, in M139 which is 3N for Chr4, the average FPKM of genes on Chr4 was 32% higher than in its 2N parent or siblings while there was no significant difference for genes on 2N chromosomes such as Chr13 and 14. Comparable data were seen for M536, which has an extra copy of Chr13 and accordingly higher mean FPKM levels of genes on that chromosome. A similar trend was observed in M738, which is 2N, 3N, and 4N for Chr4, Chr14, and Chr13, respectively, and in which the average FPKM increased progressively with copy number. Note that the mean FPKM of genes on Chr4 in M738 appears less than strain 1306 in Fig 7D, but this is an artifact of normalization since the average ploidy of M738 was closer to 3N.

The effect of aneuploidy on gene expression was more complex than simply changing mRNA abundance in proportion with copy number. For example, while the copy numbers of Chr4 and Chr15 were 50% higher than normal in F1 strains M139 and M536, respectively, the mRNA levels of genes on those chromosomes increased by an average of only 32% and 27%, respectively. This likely reflects regulatory mechanisms such as feedback control that are independent of copy number. The chromosomal imbalances in the aneuploids also influenced genes from all locations regardless of copy number, as evidenced by the much wider dispersion ($P<10^{-6}$) of expression levels in Fig 7D for the 2N chromosomes in the aneuploid progeny compared to the diploids. This could be due to *trans*-acting regulators such as transcription factors.

## Gene rearrangements and context also influence ELPs

To help identify factors besides ploidy that contribute to transcriptional differences, we also used RNA-seq to compare the two diploid parents, strains 1306 and 618. Previous studies of effectors in *Phytophthora* demonstrated that gene content and sequence can vary between isolates [25,48]. To shed light on other factors underlying ELPs, we focused on genes

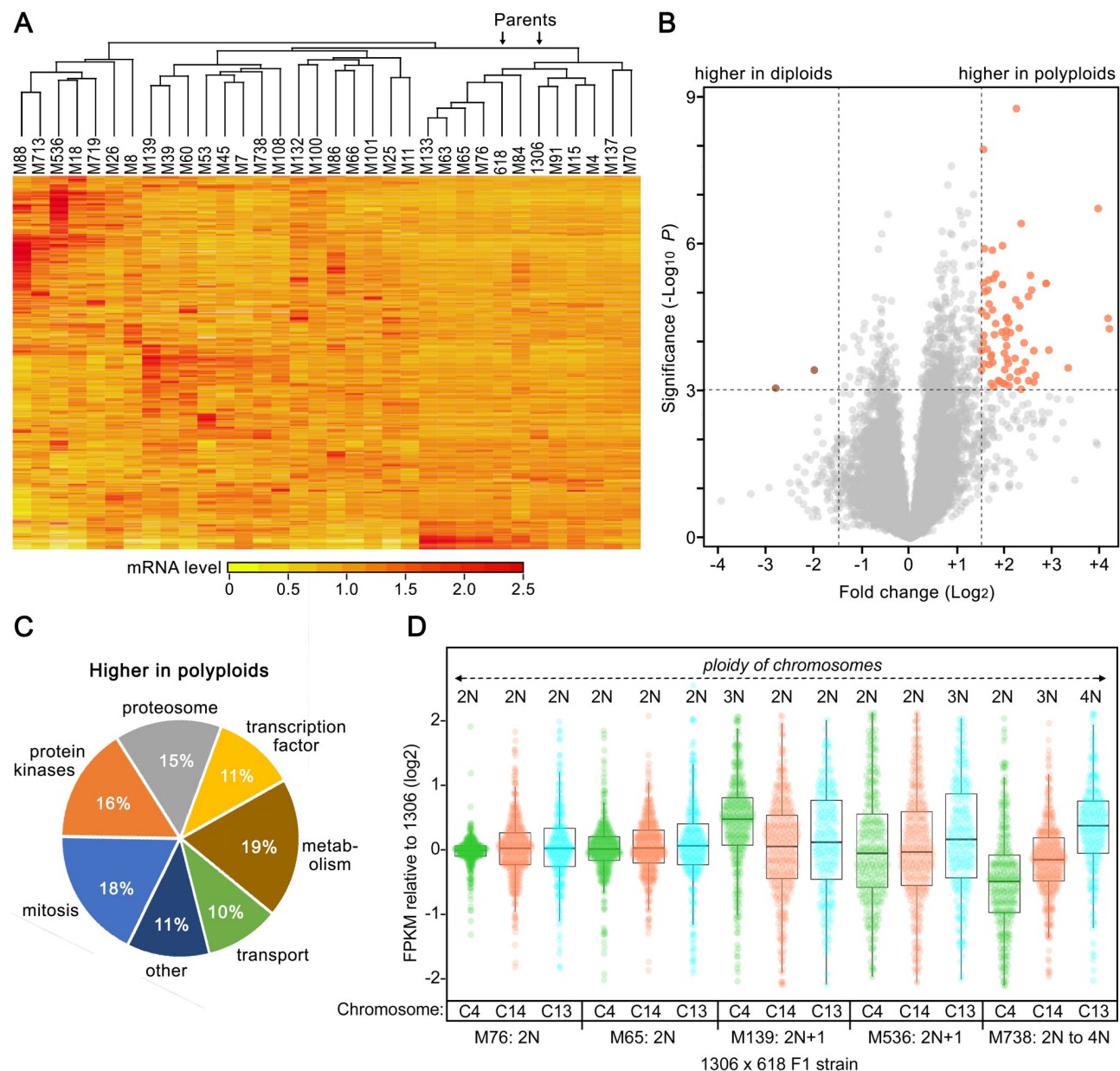

**Fig 7. Diversity of transcriptome profiles in strains 1306, 618, and their progeny. A**, Heatmap of RNA-seq data from mycelia, subjected to hierarchical clustering on both axes. Values are based on two samples per strain and are per-gene normalized to a mean of 1.0. The two parents are marked by arrows. **B**, Volcano plot comparing mRNA levels in mycelia of diploid and polyploid F1 strains. **C**, Functional classification of genes upregulated in polyploids. More details are provided in S3 and S4 Tables. **D**, Effect of chromosome dosage on gene expression. Indicated are FPKM ratios, compared to the diploid 1306, of genes on Chr4, Chr14, and Chr13 (C4, C14, and C13, respectively). Values are shown for diploid F1 strains M76 and M65, 2N+1 aneuploids M139 and M536, and F1 M738 in which chromosome numbers vary from 2N to 4N.

having ≥98% nucleotide identity between 1306 and 618, and FPKM values ≥2.0 in at least one strain to reduce noise. Fitting these criteria were 13,011 genes (Fig 8A). Of these, 1,320 exhibited >2-fold differences in mRNA levels between the strains based on an FDR threshold of 0.01. Exhibiting >100-fold differences were 164 genes, with 66 genes being expressed only in strain 1306 and 90 genes only in strain 618.

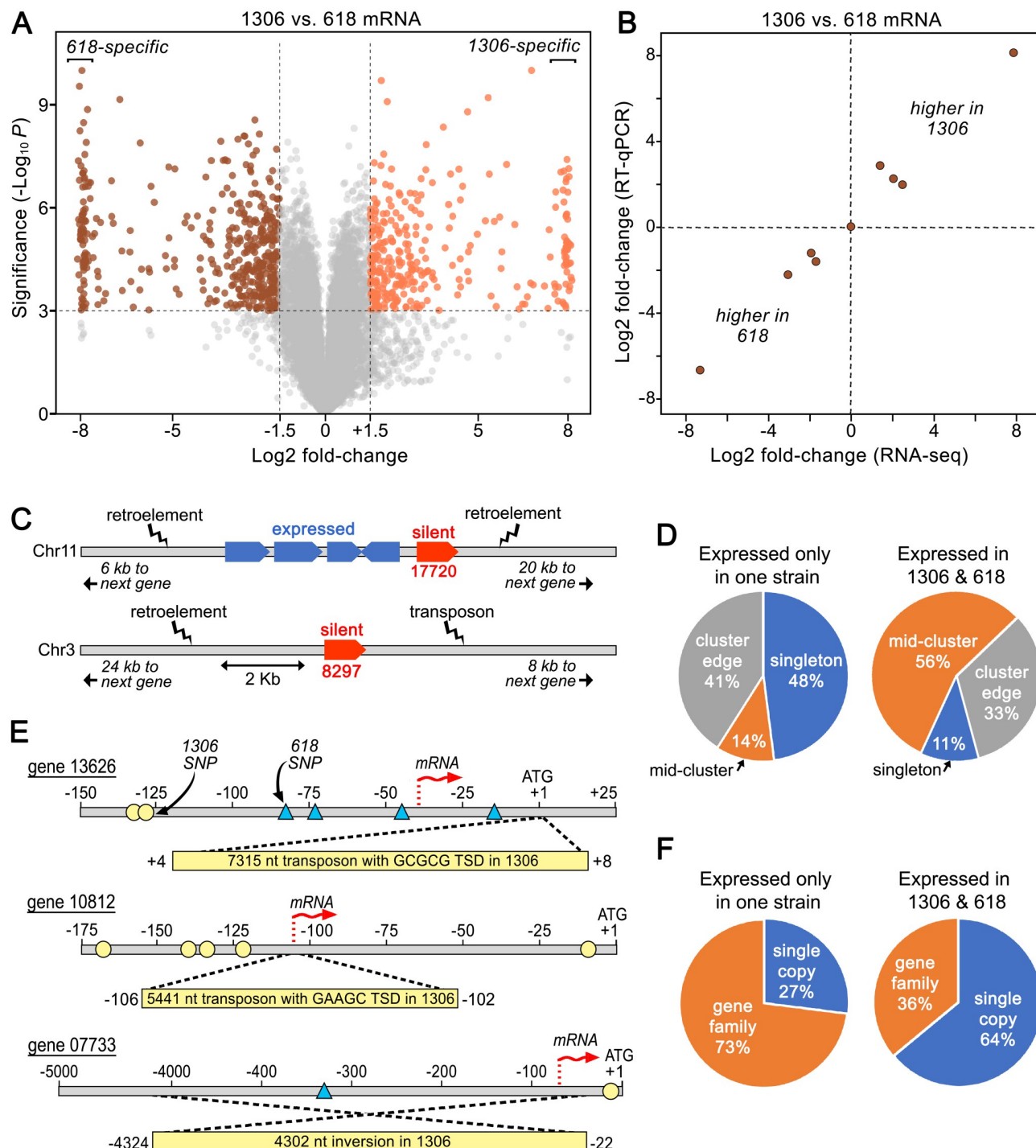

**Fig 8. ELPs between strains 1306 and 618. A**, Volcano plot from RNA-seq of mycelia. To save space, log2 fold-change differences >8.0 are mapped as 8.0, using slight off-sets to minimize overlaps. **B**, Validation of ELPs, comparing results from RNA-seq and RT-qPCR. **C**, Genomic context of two representative genes expressed in 618 but not 1306 (in red); genes in blue have similar mRNA levels in both strains. Gene 17720 resides on the edge of a gene cluster while gene 08297 is singleton. **D**, Genomic context of genes expressed in one or both strains. Genes are defined as occurring within gene clusters (mid-cluster), on the edge of a cluster, or as singletons (>2.5 kb from another gene). **E**, Examples of events disrupting expression in strain 1306. For genes 13626 and 10812, transposon-like sequences flanked by target site duplications (TSD) resides near the 5' end of the gene in strain 1306. For gene 07733 in strain 1306, upstream sequences are inverted relative to 618. SNPs in 1306 and 618 relative to the T30-4 reference genome are marked with circles or triangles, respectively. **F**, Fraction of genes that are expressed only in 1306 or 618, or in both strains, that are single-copy or belong to families.

To confirm these differences, RT-qPCR was performed against nine genes showing ELPs between strain 1306 and strain 618, using primers designed to perfectly match their targets in both strains. This used RNA from cultures independent of those used for the RNA-seq analysis. The results from RNA-seq and RT-qPCR were well-correlated (Fig 8B).

No GO, INTERPRO, or KEGG terms were over-represented among the genes exhibiting ELPs, including those showing strain-specific transcription. No category of effector was significantly over-represented, but most would have been excluded from our study since we focused on genes expressed in mycelia grown on artificial media, while many effectors are mainly transcribed in spores or *in planta* [9,26,49].

To identify the basis of the ELPs, we searched for polymorphisms in promoters of the 66 genes expressed in strain 1306 but not strain 618. SNPs or small indels were often found within 400 nt upstream of the start codon, an interval in which past studies have suggested include most transcription factor binding sites [50]. Whether these differences affect the binding of transcription factors can only be hypothesized in the absence of functional data. However, polymorphisms were sometimes not detected, implicating mechanisms involving *trans*-acting features such as transcription factors, heterochromatin spreading from flanking regions, or strain-specific microRNAs. A search of our database of small RNAs did not find any associated with genes showing diminished expression. However, the genes showing strain-specific expression were more likely to reside as singletons or on the border of gene clusters, flanked by repetitive and presumably heterochromatic DNA (Fig 8D). Examples of two such genes are in Fig 8C. Gene 17720, which is expressed only in strain 618, is located close to retroelement-like sequences on one edge of a gene cluster (note that genes in this paper are referred to by the numbers of their orthologs in the Broad Institute database, without the "PITG" prefix). Also expressed solely in strain 618 is gene 08297, a singleton flanked by retroelement and DNA transposon-like sequences.

We also detected rearrangements that would be expected to result in transcriptional quiescence of genes. Three examples which lacked detectable expression through the coding sequences of genes in strain 1306 are shown in Fig 8E. For gene 10812, a 5.4 kb transposable element flanked by a 5-nt target site duplication was found 102 nt upstream of the predicted coding sequences in strain 1306, about 4-nt downstream of the transcription start site estimated for strain 618 from alignments of RNA-seq reads. For gene 13626, strain 1306 contained a 7.3 kb transposable element flanked by a 5-nt target site duplication 46-nt downstream of the transcription start site determined for strain 618. A different type of rearrangement was observed for gene 07733. In that case, a 4.3 kb block of sequences upstream of the gene was inverted in strain 1306 compared to strain 618, moving the transcription start site active in strain 618 away from the gene in strain 1306. These rearrangements appear to be recent events since they were not observed in other sequenced strains including T30-4, 88069, KR-1, and RC1-10 [51,52].

When all of the silent genes in 1306 and 618 were examined, we discovered that overall they were much more likely to belong to a family (defined by OrthoMCL [53]) than those expressed in both strains ($P < 10^{-4}$; Fig 8F). This is logical since redundancy would minimize the likelihood that a major ELP affecting one gene in a family would be deleterious. To determine how often major ELPs occurred within families, we used the T30-4 reference genome to identify a representative group of families (n = 60) having four to eight paralogs with functions besides RXLRs and CRNs. When the relative expression of their orthologs in strains 1306 and 618 in mycelia were compared, notable differences were observed within three of the families. In one family, this was attributable to a deletion of one paralog from strain 1306. In the other two cases coding and promoter sequences were fairly well-conserved between the strains although

some SNPs in promoters were present. Consistent with the results described above, these silent genes resided at the edge of gene clusters.

## Metalaxyl resistance maps to the right arm of Chr3

To further explore the utility of our chromosome-scale assembly, it was also used to identify genomic regions associated with insensitivity to the oomyceticide metalaxyl. To allow this, the growth of 82 sequenced progeny of the 1306 × 618 cross were scored in the presence or absence of the compound (Fig 9A). A semicontinuous response consistent with a quantitative trait was seen (Fig 9B). We previously reported similar results [54] and concluded that 618 was heterozygous for a major resistance locus *(Mex/mex)* while 1306 was a sensitive homozygote *(mex/mex)*, with variation within each genotypic class determined by loci of minor effect. Based on the current study, we infer that some variation is also attributable to ploidy differences affecting copy numbers of the genes.

The progeny were subjected to QTL analysis using 55,345 and 28,771 pseudo-testcross SNP markers from parents 1306 and 618, respectively. Resistance mapped to an interval at the right end of Chr3 with a peak LOD score of 11.3 (Fig 9C, top). A moderate peak (LOD = 4.6) was identified on Chr13. Minor peaks were also found using markers from the 1306 parent, such as one on the left end of Chr10 (LOD = 4.4).

A detailed analysis of Chr3 is shown in Fig 9D. Markers from the right side of the Chr3A haplotype of 618, which contains the resistant *Mex* allele, were linked to insensitivity. Markers from Chr3B haplotype, which bears the sensitive *mex* allele, were linked to sensitivity. The zone of peak linkage started around position 15.7 Mb and extended to the right end of Chr3. A few markers in the 16.7–16.8 Mb region showed some recombination with the trait, either defining a right boundary of the *Mex*-bearing interval or an assembly error.

About 130 genes reside to the right of the 15.7 Mb location on Chr3. None encode any of the 12 subunits of RNA Polymerase I, which had been proposed to be the target of metalaxyl based on studies that suggested that it affects rRNA synthesis [55]. However, a gene encoding rRNA biogenesis protein RRP5 is at 16.1 Mb. This is a strong candidate for the major determinant of insensitivity since a recent GWAS study in *Phytophthora capsici* also associated *RRP5* with resistance to mefenoxam, an isomer-specific formulation of metalaxyl [56]. There are 21 nonhomologous SNPs between the putative resistant and sensitive alleles of this gene.

## Rearrangements of Chr3 are common and affect metalaxyl insensitivity

Intriguingly, fast-growing sectors often appeared during the propagation of *Mex/mex* progeny on metalaxyl (Fig 10A). These typically appeared after only a few days of growth. To study this in detail, we performed single-zoospore selections to obtain pure cultures from the sectors. Assays of their responses to the chemical confirmed that the fast-growing variants were more tolerant (Fig 10B).

Further analyses revealed that each fast-growing strain lacked part of the right arm of Chr3, including the region containing *RRP5*. The original F1 strains ("normal" strains such as M64N) and their "fast" derivatives (*e.g.* M64F) seemed to be mostly diploid based on measurements of Illumina read depth and allele ratios across the genome. However, the density of reads at the right end of Chr3 was reduced by half in the fast sectors (Fig 10C). Moreover, 12% to 20% of loci from that chromosome depending on the size of the deletion displayed a loss of heterozygosity signaled by a 1:0 variant ratio (Fig 10D). This involved the loss of SNP variants in *cis* to the sensitive *mex* allele. These data indicate the presence of deletions with breakpoints between positions 13.3 and 14.5 Mb, resulting in a loss of 2.4 to 3.6 Mb. A few markers at the right edge of Chr3 retained a normal read depth. While this might mean that the deletion is

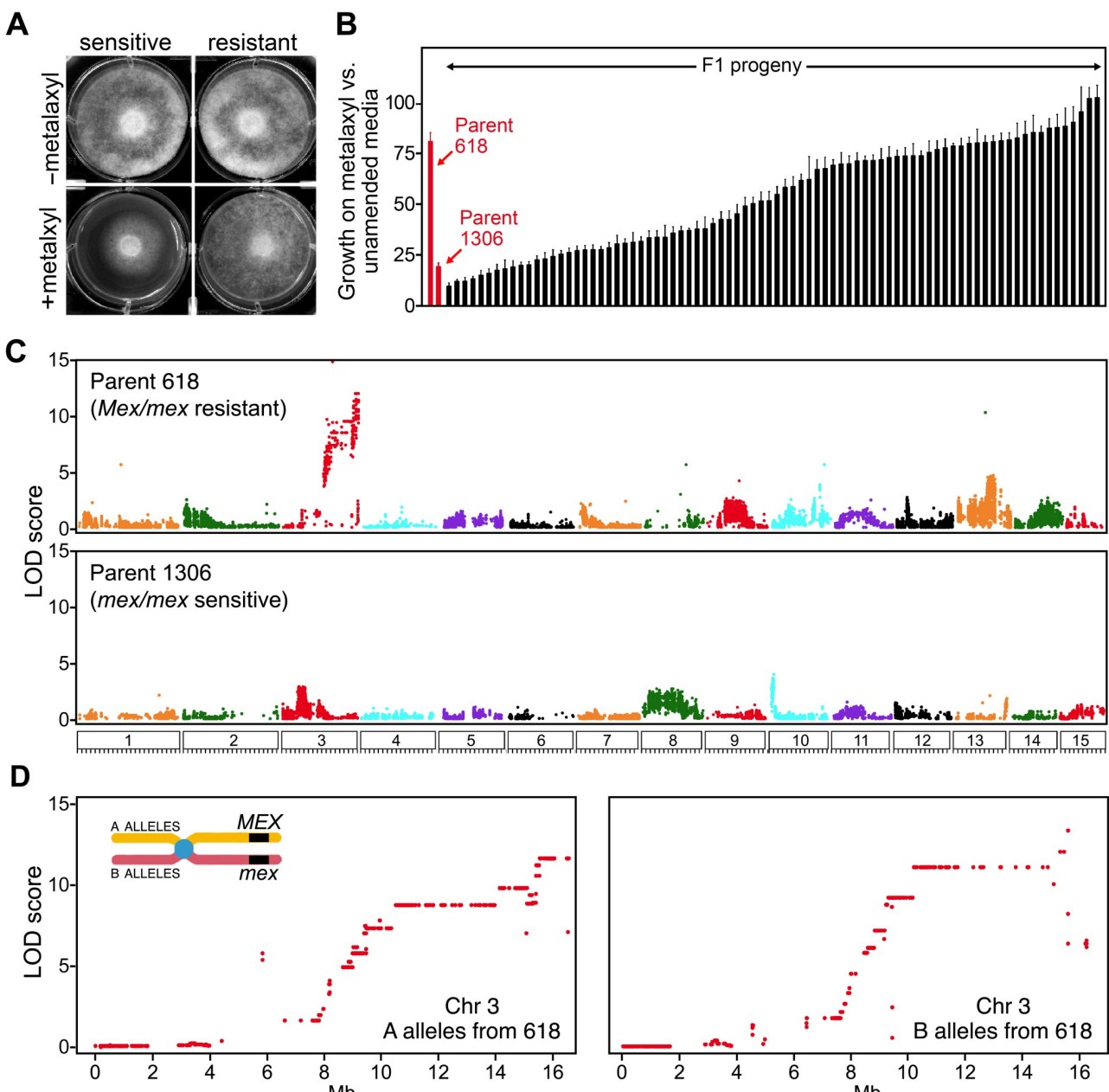

**Fig 9. Mapping loci determining insensitivity to metalaxyl. A**, growth of representative sensitive and resistant strains on unamended media (-) or media containing 0.5 μg/ml of metalaxyl (+). **B**, sensitivity of parents and their F1 progeny. Parents 618 and 1306 are inferred to be *Mex/mex* and *mex/mex*, respectively. **C**, genome-wide linkage analysis of SNP markers from 618 and 1306. **D**, detail of linkage of markers from the alternate haplotypes of Chr3 from parent 618. The chromosome illustration shows that A and B alleles are in *cis* and *trans* to the haplotype bearing the resistant *Mex* allele.

internal to the chromosome, a more plausible explanation is that these signal errors in read mapping or assembly.

That the fast-growing sectors bore deletions was verified by qPCR and high resolution melt (HRM) analysis. For qPCR, we tested DNA from the two parents, four fast-growing sectors, and their normal progenitor F1 strains using primers targeting loci inside and outside the deleted region. This confirmed that the copy number of the right side of Chr3 was reduced by half in the fast-growing sectors (Fig 10E). The HRM results were also consistent with a

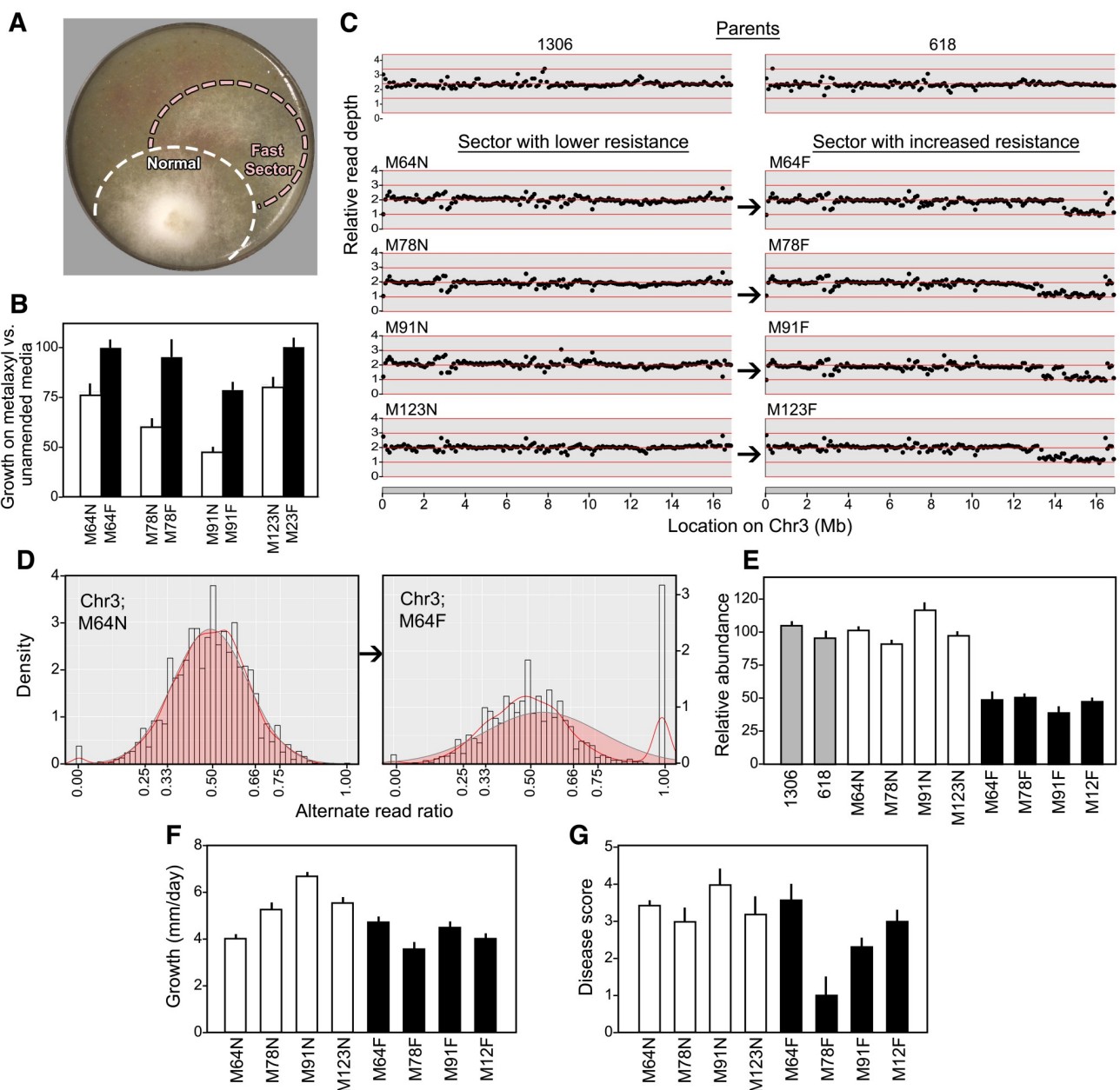

**Fig 10. Changes induced by growth on metalaxyl. A**, fast-growing sector emerging from a representative *Mex/mex* F1. **B**, relative growth rates of normal (N) and faster-growing (F) sectors from progeny M64, M79, M91, and M123. **C**, read depth along Chr3 in parents 1306, 618, and normal and fast sectors, revealing reduced copy number on the right arm of Chr3 in the fast variants. **D**, variant ratios of heterozygous SNPs along Chr3 in normal progeny M64N and its fast-growing derivative, M64F. The pileup of about 12% of markers at 1.00 reflects the loss of heterozygosity. **E**, confirmation of deletions in Chr3 using qPCR. Values indicate the ratios of the calculated concentration of the deleted region and the non-deleted region in DNA from normal progeny sectors (white), fast-growing sectors from the same progeny (black), and the parents (grey).

deletion, as they indicated a loss of heterozygosity in the fast-growing strains for a marker on the right side of Chr3, while heterozygosity persisted left of the breakpoint (S9 Fig.).

Since the deleted regions included more than 100 genes, we checked whether this deficit affected phenotypes besides metalaxyl response. The strain with the smallest deletion, M64F, grew on rye-sucrose media at a rate similar to its progenitor, M64N (Fig 10F). However, the

growth rate was reduced in M78F, M91F, and M12F, which bear larger deletions. A simple relationship was not observed between the size of the deletion and lesion development on potato tubers; while both M64F and M123F formed lesions similar in size to their undeleted progenitors, diminished lesions were made by M78F and M91F (Fig 10G). It thus appears that *P. infestans* has at least some capacity to compensate for modest changes in gene copy number.

## Discussion

We have reported a wealth of mechanisms underlying variation within *P. infestans*, using a new chromosome-scale assembly as a foundation for analysis. The assembly contains 219 Mb on the 15 predicted chromosomes, plus 28 Mb of DNA in contigs not placed on chromosomes. By comparison, the genome was previously measured at about 240 Mb by Feulgen cytophotometry [22] and flow cytometry [57]. In our study, variation within the species was detected in the form of strain-specific differences in chromosome structure, deletions arising during asexual propagation that affect chemical resistance, and major karyotype changes following sexual reproduction that altered gene copy number and mRNA levels. Transcriptional differences were also discovered between diploids, some of which could be linked to DNA rearrangements in promoters and most often affected genes belonging to families or adjacent to repetitive DNA. The latter might contribute to silencing through heterochromatin spreading [58].

We also obtained a chromosome-level view of the distribution of genes and repetitive DNA in *P. infestans*. This revealed a genome in which about 80% is a mosaic of interspersed gene- and repeat-rich domains, containing loose clusters of genes encoding RXLR and CRN effectors. While an earlier study with a more fragmented assembly concluded that RXLR genes in *P. sojae* were clustered to only a limited degree [59], their predominant grouping in *P. infestans* is clear from our assembly. Their distribution is consistent with a model in which the RXLR and CRN families expanded through unequal crossing-over, possibly mediated by flanking repetitive DNA, followed by diversification of the effector sequences. Other gene families also experienced expansions, as about 58% of all protein-coding genes bordered a gene of related sequence. The most extreme example of family expansion involved tRNA genes, which numbered more in *P. infestans* than most eukaryotes. Like effector genes, the tRNA gene families appear to have grown through unequal crossing-over but with much less divergence in sequence. It is interesting to contemplate whether the unusually large arrays of tRNA genes in *P. infestans* contribute to nuclear or chromatin organization, as shown for tRNA genes in *S. cerevisiae* [60].

We also observed multiple lengthy gene-poor regions, including apparent centromeres spanning up to 4.5 Mb each. By comparison, most centromeres are only about 200 kb in *P. sojae*, which has a smaller genome at 83 Mb [33]. In this regard, *Phytophthora* resembles plants where centromere and genome sizes are positively correlated [61]. Interestingly, the more outsized centromeres in *P. infestans* (*e.g.* Chr1, 3, 8) are close in size to those of plants with more massive genomes such as maize [62]. While centromere size has been linked to segregation fidelity in some species, a more significant factor is the abundance of kinetochore proteins assembled per centromere, which may not scale with centromere size [63,64]. This could explain why Chr12 and Chr13 of *P. infestans* were the most and least prone to become unbalanced, respectively, even though they have similarly-sized centromeres. Centromeres are defined functionally not by sequence or size but by the presence of a specialized histone H3 variant, CENP-A [65].

The frequency of polyploids and aneuploids above 2N in our *P. infestans* progeny (50%) was exceptionally high. A likely explanation for the phenomenon is non-disjunction during

meiosis, resulting in unreduced gametangial nuclei. It follows that oomycetes may lack a pachytene checkpoint [66]. Studying individual gametes is not feasible since oomycetes do not produce free gametes; the haploid nuclei are housed within antheridia and oogonia which attach during mating. We therefore can not exclude other processes such as the fusion of gametangial nuclei prior to fertilization, analogous to the situation in some fish where gametic cells can fuse [67]. Nevertheless, the frequency of polyploid progeny in *P. infestans* is at least an order of magnitude higher than the frequency of unreduced gametes in plants and animals [68,69]. It would be interesting to learn if the frequency of ploidy variation observed in our laboratory studies would resemble that from a sexual population formed naturally within a potato field, or with strains isolated from other regions of the world.

While it was remarkable to observe that about half of our F1 progeny were polyploid, the result was not totally surprising since there was a prior report of extra bands appearing during restriction fragment polymorphism (RFLP) analysis of progeny [70]. In our crosses, chromosomes from all four parents were involved, implying that such "defects" are in fact natural to the species. This distinguishes *P. infestans* from species in other taxa where polyploids are thought to arise mainly due to stress. For example, fungicides stimulate polyploid formation in the human pathogen *Cryptococcus neoformans*, which better tolerate stress than the progenitor haploids [1]. Such benefits are distinct from the long-term evolutionary advantages of polyploidy, which allows gene diversification by increasing copy number [71].

The potential of *P. infestans* polyploids to have superior stress survival may explain their prevalence in nature. Outside of a few regions where sexual reproduction is common, most strains of *P. infestans* from potato and tomato fields are triploid with tetraploids also reported [20,21,72]. Aneuploidy has not been described, however most studies describing polyploidy used techniques that would have difficulty resolving polyploids from aneuploids. These include flow cytometry [57], photocytometry [22], and microsatellite markers [73] which as shown in S5 Fig. would only sample selected chromosomes.

Besides the prospect of increasing tolerance to insults faced *in planta* such as antimicrobial phytoalexins and nutrient limitation, the increased genetic redundancy conferred by polyploidy may also help *P. infestans* (which lacks light-blocking pigments) survive damage from UV radiation. Cellular changes in polyploids such as altered nuclear surface-volume ratios may also be an advantage under some circumstances [74]. Nevertheless, polyploids did not appear to be more pathogenic than diploids in our assays, although those data are hard to interpret since pathogenicity varied widely within each ploidy class. Polyploids might also benefit from encoding a more diverse complement of effectors such as RXLRs and CRNs, which act as virulence factors [10].

In nature, polyploid strains of *P. infestans* have been proposed to revert to diploidy to facilitate sexual reproduction, with one study showing that a ploidy reduction could be stimulated by a sublethal concentration of oomyceticide [75]. Whether this reversion is truly beneficial is unclear since polyploids can yield viable progeny [20,76]. Nevertheless, the potential instability of polyploidy is consistent with the abundance of aneuploids in our progeny, which we hypothesize arose by sequential chromosome loss from a transient triploid or tetraploid state. Karyotype reduction is also common in polyploid human cancer cell lines, sometimes yielding aneuploids with transcriptional profiles that enhance tumor growth [77, 78]. Our *P. infestans* aneuploids and polyploids also exhibited distorted patterns of transcription that were not simply proportional to gene copy number. Complex (and nonadditive) transcriptional responses to polyploidy have also been described in plants [79]. In *P. infestans*, some genes involved in mitosis were upregulated in polyploids which is suggestive of a cellular struggle to deal with increased nuclear content. Such forces may lead to the occasional destabilization of polyploids within a culture or plant lesion.

Besides karyotype changes affecting ploidy, we also observed deletions involving the region containing the gene for metalaxyl resistance. The basis of resistance has been sought for decades, since metalaxyl has been useful for controlling oomycetes in many pathosystems. While the 82 progeny analyzed did not confer sufficient resolution to narrow the determinant to a single gene on Chr3, linked to the trait in our cross and present in each deleted region was *RRP5*, which had also been identified by a GWAS study in *P. capsici* [56]. This gene is needed for the maturation of ribosomal RNA, which is congruent with biochemical studies that suggested that the target of metalaxyl was RNA polymerase I or its products [55]. It is likely that the Chr3 truncations involving this locus were facilitated by the repeat content of the genome, through unequal crossing-over or a transposon burst activated by metalaxyl-induced stress. Similarly, studies in the sudden oak death pathogen *Phytophthora ramorum* found that host-induced stress caused genome instability and transposon derepression [80]. While not examined here, transposon insertions and subsequent double-strand break repair might also induce gene conversion and copy-neutral losses of heterozygosity.

Our assembly represents an improvement over existing resources for *P. infestans*. It remains a work in progress as gaps persist and 11% of sequences were not placed into chromosomes. Ours is not the first oomycete assembly to be organized into linkage groups, as this was described for the smaller 56 Mb genome of *P. capsici* [81]. Nevertheless, our combination of a genetic map, long reads, and an optical map resulted in fewer and longer scaffolds that help map QTLs, identify karyotypes and structural variants, and visualize genome organization. Further improvements might be achieved using other sequencing technologies or phasing. Comparisons to a recent assembly of the downy mildew *Pe. effusa*, the first telomere-to-telomere assembly of an oomycete [43], may also help assess the fidelity of our genetic linkage approach for joining sequence-based contigs into chromosomes. Our method assumes that *P. infestans* SNPs in linkage disequilibrium are linked physically, but this may not be the case due to factors prevalent in many organisms such as epistasis [82].

A major biological conclusion about *P. infestans* from this study is that its genome is fluid and adaptable, being capable of shifting ploidy, adjusting gene content, and altering patterns of transcription. The ease at which Chr3 deletions arose in our metalaxyl experiments, along with evidence of recent transposon activity affecting gene expression, suggests that similar processes could help *P. infestans* overcome other oomyceticides. The same mechanisms may also enable *P. infestans* to evade host resistance by eliminating avirulence proteins that are recognized by the plant immune machinery. Particularly since some avirulence loci are hemizygous [83], this may help explain how *P. infestans* has defeated most resistance genes deployed through plant breeding programs [84]. Future investigations of the biological consequences of polyploidy may also illuminate why *P. infestans* and relatives have evolved to be such successful pathogens, able to generate new and often more aggressive genotypes [85].

## Methods

### Growth and phenotyping of *P. infestans*

Strains were were maintained on rye-sucrose agar at 18˚C. These included A1 mating type isolates 1306 (isolated in 1982 from *Solanum lycopersicum* in California USA) and 6629 (isolated in 1983 from *Solanum tuberosum* in Mexico) and A2 strains 550 (isolated in 1983 from Solanum *stoloniferum* in Mexico) and 618 (isolated in 1987 from *Solanum tuberosum* in Mexico). Strains are available from the *Phytophthora* collection housed at the University of California,

Riverside. Crosses were made as described [76]. In brief, this involved extracting oospores from the interaction zone between A1 and A2 hyphae, plating the oospores on water agar, and using a needle and dissecting microscope to pick and transfer single germinating oospores to rye-sucrose agar. That oospore-derived strains were true outcrosses was confirmed by genotyping. Single nuclear-derived cultures were obtained by inducing sporangia to produce zoospores, which were purified by passage through 10 μm nylon mesh and spread on a 0.8% water agar plate. After three days, individual germinating zoospores were extracted with a needle and transferred to fresh media.

Metalaxyl sensitivity assays were performed on rye-sucrose agar containing 0, 0.5, and 5.0 μg/ml metalaxyl (technical grade, 93% active ingredient), using at least three biological replicates. Growth rates were determined from the slopes of linear regression lines fit to the data. Lesion development was assayed on ~6 cm diameter potato tuber slices (*cv*. Russet Burbank). This entailed incubating a sporangial suspension (20/μl) at 10°C for 1 hr to induce zoospore release, followed by placing a 10 μl droplet in the center of a 3-mm thick tuber slice. After incubation for 5 d in a humidified chamber in the dark, disease ratings were assigned as follows: 0, no necrosis or visible mycelia growth; 1, symptoms limited to the inoculation site; 2, lesion <1 cm in diameter; 3, lesion between 1 and 3 cm; 4, lesion >3 cm in diameter; 4.5, lesion covering the entire tuber. Each F1 strain was assayed at least twice. Differences between groups were tested for significance using ANOVA.

## DNA isolation

Genomic DNA for Illumina, Pacific Biosciences, and Dovetail sequencing was obtained by growing strains in clarified rye-sucrose broth, collecting the mycelia by filtration, and grinding in liquid nitrogen. For strains 1306, 618, 6629, and 550, the frozen powder (ca. 4 ml) was thawed in 10 ml of extraction buffer (0.2 M Tris pH 8.5, 0.25 M NaCl, 25 mM EDTA, 0.5% SDS), and extracted with 7 ml of phenol plus 3 ml of chloroform:isoamyl alcohol (24:1) by gentle rocking (50 rpm) on a rotary platform. After centrifugation at 10,000 ×*g* for 30 min, the supernatant was removed with a wide-bore pipette, and re-extracted with an equal volume of chloroform:isoamyl alcohol. The final aqueous phase was treated with 50 μl of 10 mg/ml RNAse A at 37°C for 30 min, and then the DNA was precipitated by adding 0.6 volumes of isopropanol followed by centrifugation at 10,000 ×*g* for 20 min. The pellet was rinsed with 70% ethanol, air-dried, resuspended in TE (10 mM Tris pH 7.5; 1 mM EDTA) by overnight incubation at 4°C, and subjected to field-inversion gel electrophoresis which indicated that the DNA had a mean size >75 kb. For the F1 progeny, DNA was prepared using the GeneJET Plant Genomic DNA Purification kit. DNA from both methods was quantified using the Qubit Broad Range DNA kit.

To obtain ultra high molecular weight DNA for optical map analysis, protoplasts were prepared as described [86]. These were resuspended in 1M mannitol, 0.1 M sodium citrate pH 7.0, 60 mM EDTA at 8 x10$^7$ per ml. After being held at 43°C for 30 sec, the protoplasts were mixed with an equal volume of 2% low melting agarose in the same buffer held at 43°C. The mixture was solidified in a plug mold at 4°C, and then the plugs were dislodged and transferred to a 50 ml tube bearing 2.5 ml of lysis buffer (200 μl of 20 mg/ml proteinase K, 0.5 M EDTA pH 9.3, 1% w/v sodium lauryl sarcosine, 25 μl β-mercaptoethanol). After overnight incubation at 50°C, the liquid was replaced with fresh proteinase K solution and incubated at 50°C for 2 hr. The tube was then shifted to room temperature, mixed with 50 μl of 100 mg/ml RNase A, held at 37°C for 1 hr, and then the plugs were then rinsed seven times in 10 ml of 10 mM Tris pH 8.0, 50 mM EDTA pH 8.0. Field-inversion gel electrophoresis indicated that the average size of DNA was >1 Mb.

## DNA sequencing, optical mapping, and initial assemblies

PacBio SMRT sequencing was performed on strain 1306 by the UC-Davis core facility, resulting in an unfiltered output of 24.8 Gb in 3.2 million reads, or 113-fold genome coverage. Optical map data for 1306 were developed by the Bionano facility at Kansas State University, using *Bsp*QI and *Bss*SI as nicking enzymes. A long-range linking library based on the Chicago method [87] was constructed and sequenced by Dovetail Genomics (Scott's Valley, California, USA), yielding 202 million reads.

Paired-end Illumina Hiseq reads of parents and progeny were generated by core facilities at University of California-Riverside and University of California-Davis, or by Cofactor Genomics (St. Louis, MO, USA), using PCR-free library kits. This yielded between 50 and 100-fold coverage for strains 1306, 618, 6629, and 550, and 15 to 30-fold coverage for the F1 progeny.

We described previously the use of the tool Novo&Stitch [27] to develop a preliminary assembly for 1306, which we named Stitch5. This reconciled the optical map with outputs of the ABruijn, CANU, and Falcon assemblers [88–90], and included polishing using Quiver and PILON [91], and the detection and splitting of chimeric contigs by Chimericognizer [92]. The data was then subjected to additional refinement including a run by Dovetail Genomics through their HiRise pipeline. This resulted in an assembly named Stitch7. This was run through BUSCO 5.2.2 [29] using the Augustus option.

## Chromosome-scale assembly

These and other analyses were performed on the UC-Riverside High Performance Computing Cluster, the Galaxy Platform [93], or dedicated local workstations. To identify markers for linkage analysis, Illumina reads were trimmed using Sickle (github.com/najoshi/sickle) using default settings. The reads were then aligned to Stitch7 using BWA-mem [28] and variants called using GATK best practices [94], retaining all SNPs and indels ≤3 nt. Markers were generated to resemble a pseudo-testcross (Aa × AA or AA × Aa) and filtered to include those showing Mendelian segregation ($X^2$ ≤0.05). To reduce miscalls resulting from low sequence coverage, particularly homozygous calls made when heterozygosity is the true state, we imputed correct genotypes when possible. To enable this, genotypes of each parent were split into two VCF files consisting of markers that were heterozygous in parent1 and homozygous in parent2, and *vice versa*. The VCF files were then subsampled with vcftools [95] to ensure at least a 5 kb gap between markers, resulting in 23,661 markers between the four parents. Phases were then determined by Beagle 4.0 [96], using a window of 1000 and an overlap of 300. For parents 1306, 618, 6629, and 550, we removed 758, 567, 246, and 271 markers that showed unexpected segregation ratios, respectively, leaving a total of 22,010 phased markers. Genotype-corrector [97] was then used to identify miscalls and impute more parsimonious genotypes using default parameters and an F2 population type, excluding contigs with fewer than 15 markers. This was applied to 1,380,769 genotypes across the crosses, resulting in 15,888 corrections mostly (85%) involving changes from homozygous to heterozygous states. After incorporating markers modified by Genotype-corrector and adding some additional contigs from the FALCON assembly, 6,993 markers were identified and passed to JoinMap 5.0 (Kyasma, Wageningen, Netherlands).

JoinMap generated independent genetic maps for each haplotype of the four parents, resulting in eight genetic maps. Initial genetic sizes were bloated due to aneuploidy and errors not modified by Genotype-corrector. Consequently, a first round of corrections addressed miscalls due to aneuploidy, which can force genotypes to exist as completely hetero- or homozygous across a contig regardless of haplotype phase. A second round focused on miscalls at the edges of contigs. A third round removed miscalls inside small contigs which had been excluded

previously from Genotype-corrector. A final round eliminated 770 markers placed more than 30 cM from the nearest marker. The eight polished genetic maps, which incorporated 6,166 markers, were then assembled into chromosomes using ALLMAPS [98], adding 100 Ns between nonoverlapping contigs. All Illumina sequencing fastq files were subsequently remapped to the chromosome-scale assembly, and all heterozygous polymorphisms for downstream analyses were recalled via GATK [99] as executed with Stitch7.

## Genome annotation

Prior to annotation, repetitive DNA were identified and masked using RepeatMasker with a custom library built by RepeatModeler. Some investigations of repetitive DNA post-annotation also used the GIRI Repbase [100]. Annotation was performed with the Funannotate pipeline [35] using evidence from previously predicted protein sequences from *P. infestans*, *P. parasitica*, *P. sojae*, and *Pythium ultimum*, plus mRNAs from strain 1306. The latter included 110,667 transcripts assembled with PASA [101], using 346 million Illumina RNA-Seq reads representing multiple developmental and infection stages [26,49,102]. In the pipeline, the transcripts, proteins, and RNA-seq reads were aligned to the 1306 genome using Minimap2 [103], Exonerate [104], and bwa [28], respectively, using a mapping quality threshold of 20. These alignments were used to train Augustus [105] for *de novo* gene predictions, which in turn were employed to train Snap and GlimmerHMM [106]. Predictions from these plus GeneMark [107] were then passed to EvidenceModeler [101] to generate comprehensive gene models. Combined with tRNAscan-SE [41], this pipeline identified 19,981 protein-coding and 11,056 tRNA genes. Genes in 1306 that belong to the RXLR and CRN effector families were identified using EffectR [108]. Since many effector genes are small and not easily identified by conventional pipelines, we lifted over additional RXLR and CRN genes annotated in the T30-4 assembly to 1306 using Flo [109]. Genes were considered as being part of a cluster if they were within 2.5 kb of another gene.

## Karyotype assessments

This integrated results from sequencing read depth analyses with allele read ratio data, based on Illumina reads. For the former, normalized read depths were calculated using bin-by-sam [110].

Two datasets were constructed to calculate allele read ratios in the progeny. The first consisted of all heterozygous, biallelic SNP and indel polymorphisms. This was generated from genotypes scored as heterozygous by GATK UnifiedGenotyper, filtered to include those with a read depth >8 and indels <6 nt. The second dataset included homozygous polymorphisms between the parents (*e.g.* AA in 618 and TT in 1306). Since 82 F1 strains from the 1306×618 cross had been sequenced, polymorphisms were excluded if the number of F1s with called heterozygous genotypes fell below 70, resulting in 38,039 markers. Similar datasets were generated for the 32 sequenced F1 strains from 6629 × 550. After filtering for variants heterozygous in at least 28 of the 32 progeny, 38,034 and 40,017 homozygous alternate variants were identified from strains 550 and 6629, respectively.

Allele read ratios were generated by dividing each alternate read depth by the total read depth per variant. For heterozygous variants originating from a locus with a copy number of two (as in a diploid), allele ratios should be normally distributed around 0.50. Variants at a three-copy locus (as in a triploid) would exhibit a bimodal distribution with peaks at ratios of 0.33 and 0.66, while variants at a four-copy locus would have trimodal (0.25, 0.50, and 0.75) or unimodal (0.5) distributions. Regions of variants showing extensive loss of heterozygosity (LOH), either through chromosome nondisjunction, truncations and duplications, or

copy-neutral LOH, will not show normal distributions, and will instead shift to highly skewed ratios, including 0.0 or 1.0 when only the reference or alternate allele is present, respectively. The variant distributions were analyzed and plotted using ggplot2 [111] and modeled using fit-distrplus [112] and mixtools [113].

## RNA-seq

RNA was extracted using the Spectrum Plant Total RNA (Sigma, St. Louis, MO USA) from mycelia grown in rye-sucrose broth, with four samples per parent and two per F1 strain. RNA quality was assessed by electrophoresis as well as using an Agilent 2100 Bioanalyzer. Samples with an RNA Integrity Number (RIN) >7.0 were sequenced on the Illumina platform at Cofactor Genomics (St. Louis, MO USA) to an average of 33.6 million 75-nt single-end reads. Using the SystemPipeR workflow [114], these were aligned and mapped to *P. infestans* gene models using Bowtie version 2.2.5 and Tophat version 2.0.14, allowing for 1 mismatch [115,116]. Expression and differential expression calls were made with edgeR [117]. False discovery rates were calculated based on the Benjamini-Hochberg method. Since an assembly and gene models were not available for strain 618, models from the Broad Institute T30-4 reference genome were used in order to minimize bias when comparing strains 1306, 618, and their progeny. These gene numbers are listed in Results without the "PITG" prefix. To exclude genes with unreliable expression calls, the data in Fig 7A only include genes with FPKM >1.0 in >50% of the samples. Similarly, studies of expression differences due to ploidy (Fig 7B–7D) were trimmed to only include genes with FPKM >2.0. To avoid false calls of ELPs resulting from sequence divergence, comparisons of strains 1306 and 618 (Fig 8) were restricted to genes showing >98% nucleotide conservation between the strains, measured by aligning gene models against Illumina reads. A few genes exhibiting strain-specific expression based on RNA-seq were eliminated from the analysis based on having unrealistic gene models; these typically had unusually large introns and lacked annotated functions or orthologs in other *Phytophthora* spp. Tests for functional enrichment were performed using ShinyGo [118].

## Small RNA analysis

RNA was extracted from 1306 hyphae using TRI reagent (Molecular Research Center, Cincinnati, OH USA), and resolved by electrophoresis on a 15% polyacrylamide-7M urea gel. The 18–30 nt region was excised and small RNAs recovered by soaking the smashed gel overnight in 0.3 M NaCl, followed by precipitation with ethanol. This was then used to generate a library using the Small RNA Library Prep Kit (New England Biolabs). Using a HiSeq2500, 31.6 million single-end reads (max = 50 bp) were obtained and submitted to Trim Galore (https://github.com/FelixKrueger/TrimGalore) to remove adaptors and low-quality reads. Reads were mapped to the *P. infestans* genome using Bowtie, filtered to exclude reads <18-nt, and for some applications also filtered to exclude reads with map scores <20.

## QTL analysis of metalaxyl resistance

Trimmed Illumina files from the crosses were aligned to the *P. infestans* chromosomes from 1306 using bwa-mem, and variants called by GATK HaplotypeCaller [99], both using default settings. Briefly, progeny were called in GVCF mode as a cohort and filtered according to its suggested hard filtering parameters, excluding indels longer than 6 bp. Variants were extracted from the cohort VCF file in a pseudo-testcross manner, thus representing SNPs as heterozygous in one parent, *i.e. Aa × aa* or *aa × Aa*. Markers were excluded if its total number of heterozygous genotypes fell below 30% or above 70% of total calls. Since the reference sequence was from 1306, no homozygous alternative calls were expected to be found, and thus alternate

(strain 618-derived) allele frequencies below 0.18 and above 0.32 were excluded. Ultimately, 28,771 high quality mapping markers for strain 1306 and 53,345 for 618 were identified. In addition, nonsynonymous variants were predicted using systemPipeR annotation features [114], with 12,061 originating from parent 618.

QTL analysis of these markers and the level of resistance to metalaxyl was performed using a custom mapping script provided by Prof. Shizong Xu [119]. This incorporates multiple polygenic covariance structures, partitioning the total genetic variance into additive, dominance, additive × additive, dominance × dominance, additive × dominance, and dominance × additive variance kinship matrices to help control the genetic background of the QTL mapping model. For Chr3, we also phased the markers from parent 618 using Beagle 4.0 [120], with miscalled genotypes corrected with Genotype-corrector [97]. Logarithm of Odds (LOD) scores were plotted with karyoploteR [121].

### qPCR and HRM analysis

qPCR was performed in a Bio-Rad CFX96 using the primers in S6 Table with the Dynamo HS SYBR Green qPCR Kit (ThermoFisher, Waltham, MA USA). Templates were either genomic DNA or cDNA prepared using the Maxima First-Strand cDNA Synthesis kit (ThermoFisher) with RNA treated with RQ1 DNAse (Promega, Fitchberg, WI USA). Reactions were performed in triplicate in a volume of 20 µl using 10 ng of template, 0.2 µM of each primer, and 40 cycles of 10 sec at 95˚C, 30 sec at 55˚C, and 30 sec at 72˚C. For RT-qPCR, constitutive gene 09862 (GenBank accession NW_003303742; [102]) was used for normalization, and parallel reactions were performed using RNA without reverse transcriptase treatment as a negative control. HRM assays were performed using the same cycling conditions with the primers shown in S6 Table, but using the Bio-Rad Precision Melt Supermix kit. Melt curves were generated using Bio-Rad Precision Melt Analysis software.

## Supporting information

**S1 Table. Current assemblies of *Phytophthora infestans*.**
(PDF)

**S2 Table. Chromosome and linkage group sizes.**
(DOCX)

**S3 Table. Genes differentially expression between polyploids and diploids.**
(PDF)

**S4 Table. Over-represented GO, KEGG, and Interpro terms associated with ploidy level.**
(XLSX)

**S5 Table. Genes with expression level polymorphisms between strains 1306 and 618.**
(XLSX)

**S6 Table. Primer sets.**
(DOCX)

**S1 Fig. Chromosome assemblies.** Indicated are genetic maps of each parent (left) and plots of genetic versus physical distance in each parent (right), as described in the legend to Fig 3.
(TIF)

**S2 Fig. tRNA genes in the 1306 genome. A**, genomic distribution of tRNA genes, with red bars representing the fraction of tRNAs corresponding to each codon per chromosome. The total number of genes per family on the chromosomes is shown on the right. The analysis

excluded 1,292 genes predicted to encode pseudogenes. **B**, copy numbers of representative families predicted from the assembly (blue) and calculated by read depth analysis (red). Codons bound by the tRNAs are indicated. **C**, relationship between gene copy number and the frequency of the corresponding codons in genes.
(TIF)

**S3 Fig. Diversity in tRNA families in the Oomycota. A**, number of tRNA genes predicted per species plotted as a function of genome size. The species belong to the genus *Phytophthora* (*Ph*. prefix, red-filled circles), the *Pythium-Globisporangium* group (*Py*. and *Gl*., green), the downy mildew group including *Bremia*, *Peronospora*, *Plasmopara*, and *Sclerospora* (*Br.*, *Pe.*, *Pl.*, *Sc.*, cyan), and *Aphanomyces* (*Ap.*, black). **B**, relative frequency of tRNA genes corresponding to the 20 amino acids plus selenocysteine (Sec) in each genome. This analysis was limited to the largest assembly based on long reads available in NCBI Genome, except for *P. infestans* which includes three assemblies. Accession numbers of the assemblies are GCA_001482985.1, *Ph. nicotianae*; GCA_001887855.2, *Sc. graminicola*; GCA_001974925.1, *Pl. viticola*; GCA_003730235.1, *Py. guiyangense*; GCA_003956735.1, *Ph. ramorum*; GCA_004359215.2, *Br. lactucae*; GCA_004380875.1, *Pl. halstedii*; GCA_005966545.1, *Py. oligandrum*; GCA_006386115.1, *Gl. (formerly Py.) splendens*; GCA_007655245.1, *Ph. citricola*; GCA_009848525.1, *Ph. sojae*; GCA_011316315.1. *Ph. infestans RC1-10*; GCA_011320135.1, *Ph. betacei*; GCA_011800735.1, *Pe. destructor*; GCA_012552325.1, *Ph. infestans KR2*; GCA_012656075.1, *Ph. hibernalis*; GCA_012656105.1, *Ph. syringae*; GCA_014706105.1, *Ph. quercina*; GCA_014706135.1, *Ph. tubulina*; GCA_014706215.1, *Ph. versiforme*; GCA_016169925.1, *Ph. vignae*; GCA_016618375.1, *Ph. capsici*; GCA_018691715.1, *Ph. cinnamomi*; GCA_019828595.1, *Ap. cochlioide*s; GCA_020226015.1, *Ph. colocasiae*; GCA_021491655.1, *Pe. effusa*; and GCA_023611945.1, *Ph. pini*.
(TIF)

**S4 Fig. Variation within tRNA gene arrays. A**, Sequence variation within array of genes encoding tRNA^leu (AAG anticodon) on Chr9. The genotype of each copy (ellipses) is color-coded according to the sequences in the alignment, in which bases diverging from the genome-wide consensus are colored. The hashmark on the chromosome represents a gap in the assembly. **B**, same as panel A but showing genes encoding tRNAmet (CAT anticodon) on Chr8 and Chr12. **C**, same as panel A but for a representative portion of the array encoding tRNA^trp (CCA anticodon) on Chr1. As illustrated below the chromosome, this tRNA occurs as part of a trimeric repeat encoding tRNA^ser, tRNA^trp, tRNA^thr. Variation within the ser- and thr tRNAs is not shown.
(TIF)

**S5 Fig. Categories of repetitive DNA in the genome.** Indicated are the number and total size in nt of LTR retrotransposons, DNA transposons, LINES, SINESs, simple repeats and satellites, genes encoding small RNAs (including tRNA, rRNA, snRNA, scRNA, srpRNA), and elements not classified by RepeatMasker.
(TIF)

**S6 Fig. Mapping mRNAs along chromosomes.** Shown for representative chromosomes are density graphs of mRNA (red), displayed as RPKM for each 50-kb bin. The green and blue graphs are densities of genes and repetitive DNA taken from Fig 2.
(TIF)

**S7 Fig. Location of SSRs and markers for population genetics. A**, Total SSRs in the genome are indicated by red lines. SSRs unplaced on chromosomes, which represent 9.2% of the total,

are not shown. As in Fig 2, the alternating white and grey blocks on each chromosome represent contigs from the Stitch7 assembly. Marked above the chromosomes are the locations of SSR markers used commonly in population genetics studies (black text) and RG57 ("57" in blue text). The SSR locations are based on sequences described by Lees *et al.* [23] and Li et al [122]. The RG57 locations were detected by searching the genome with GenBank accession JN160727. **B**, distribution of repeat unit sizes in SSRs.
(TIF)

**S8 Fig. Distribution of sRNAs. A**, indicated for each chromosome are the densities of genes and repeated DNA (from Fig 2), and the number of sRNAs mapped per 50 kb window. The blue bar graph represents sRNAs not filtered for mapping quality (thus retaining sRNAs mapped to repetitive sequences), while the orange and grey bars represent uniquely-mapping sRNAs in the 21-nt and 25-nt categories, respectively. **B**, size distribution of small RNAs. **C**, small RNAs mapped to a representative 60-nt region from Chr15, showing positions of sequences resembling retroelements (red), DNA transposons (gold), and genes (green).
(TIF)

**S9 Fig. Confirming copy number variation by high resolution melt analysis.** Melt profiles were developed for amplicons within and outside the region of Chr3 proposed to be deleted in the fast-growing sectors. The amplicons were generated using primers flanking sites that were polymorphic between the parents but homozygous within each parent. Thus, melt analysis should distinguish the homozygous (including monoallellic) and heterozygous (biallelic) states. The top two graphs show that melt profiles of an amplicon from the right end of Chr3 from normal and fast-growing sectors are distinct, with the fast-growing sector matching the homozygous parent. The same result was obtained for each pair of fast and slow-growing sectors from the four F1 strains. The lower two graphs, obtained for an amplicon from an undeleted region on Chr3, are consistent with its heterozygosity in both sectors.
(TIF)

## Acknowledgments

We thank Audrey Ah-Fong, Luca Comai, Thomas Girke, Zhenyu Jia, Weihua Pan, Jolly Shrivastava, Jason Stajich, and Shizhong Xu for sharing advice and/or programs. This work was supported by grants to HSJ from the USDA National Institute of Food and Agriculture and National Science Foundation and to SL from the National Science Foundation.

## Author Contributions

**Conceptualization:** Howard S. Judelson.

**Formal analysis:** Qihua Liang, Stefano Lonardi, Howard S. Judelson.

**Funding acquisition:** Howard S. Judelson.

**Investigation:** Michael E. H. Matson, Howard S. Judelson.

**Supervision:** Stefano Lonardi, Howard S. Judelson.

**Writing – original draft:** Michael E. H. Matson.

**Writing – review & editing:** Qihua Liang, Stefano Lonardi, Howard S. Judelson.

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
