## [Decision Letter · Decision Letter 0]

5 Aug 2022

Dear Dr Judelson,

Thank you very much for submitting your manuscript "Karyotype variation, spontaneous genome rearrangements affecting chemical resistance, and expression level polymorphisms in the phytopathogen Phytophthora infestans revealed using its first chromosome-scale assembly" for consideration at PLOS Pathogens. As with all papers reviewed by the journal, your manuscript was reviewed by members of the editorial board and by several independent reviewers. The reviewers appreciated the attention to an important topic. Based on the reviews, we are likely to accept this manuscript for publication, providing that you modify the manuscript according to the review recommendations.

Your manuscript has now been seen by four reviewers. You will note that reviewer 1, 3, and 4 are very supportive and highlight the overall quality and depth of the presented data. In contrast, reviewer 2 is concerned about the descriptive nature of the presented work, which I acknowledge as to be often the case in genome-focused manuscripts. Nevertheless, a near complete telomere-to-telomere of a major oomycete plant pathogen and an in depth description of its genomic features, especially ploidy shifts and LOH linked to fungicide resistance, can be considered a significant contribution to the field, and will open up new research in the future. 

I would like to invite you to address reviewers' recommendations attached to this email (note: the report of reviewer 2 is in an attachment). Next to comments and suggestions aimed to increase clarity and readability (reviewer 1, 3, and 4), I would like to particularly draw your attention to few suggestions raised by review 1, 2 and 4. Reviewer 2 and 4 mention that the used parental strains and their offspring might not be typical for natural P. infestans populations. As proposed by reviewer 4, I would suggest to acknowledge the nature of the strains and to discuss the implications in the revised manuscript in more detail. Reviewer 4 also suggests to include a discussion, and potential comparison, of the new assembly to the telomere-to-telomere assembly of P. effuse (Fletcher et al. 2022). Reviewer 1 suggests an additional computational analysis of the tRNA to study their evolution in oomycetes, which is an easy analyses worth to include. Reviewer 2 and 4 both mention that additional analyses that would link ploidy changes to virulence would be particularly impactful - if already available, this data could indeed be included. I also agree with reviewer 4 that generating novel data to address this is beyond the scope of the manuscript, and thus I leave this decision to the authors' discretion.

Sincerely,

Michael F. Seidl

Guest Editor

PLOS Pathogens

Bart Thomma

Section Editor

PLOS Pathogens

Kasturi Haldar

Editor-in-Chief

PLOS Pathogens

orcid.org/0000-0001-5065-158X

Michael Malim

Editor-in-Chief

PLOS Pathogens

orcid.org/0000-0002-7699-2064

Reviewer's Responses to Questions

Part I - Summary

Reviewer #1: In this study, Matson et al provide an updated assembly for the late-blight pathogen Pythophthora infestans and use it to explore the genetic bases of phenotypic variation in this species. Not only do the authors provide a very useful and significant genomic resource for the community, but they also use an effective combination of bioinformatic tools, classical genetic mapping, progeny crosses and phenotyping to answer compelling questions regarding pathogen biology and genome evolution. Among their many interesting findings are those regarding a possible expansion of tRNA genes within P. infestans and unusual patterns in large-scale karyotype evolution among progeny of laboratory crosses.

I commend the authors on a well-written and impressive manuscript. I have no major comments, just several suggestions where minor improvements could be made to the manuscript (see below). In particular, I think that readers would find it useful if the authors measured variation within and between tRNA families given their unusual expansion patterns, and if the authors clarified exactly how they defined a “gene cluster”.

Reviewer #2: (No Response)

Reviewer #3: This manuscript by Matson et al describes a chromosome level genome sequence assembly for the oomycete plant pathogen, Phytophthora infestans. This assembly appears to be largely complete, spans 247 Mb, features improved gene prediction, and transcripts, effectors, repeats, and sRNAs are mapped onto the assembly. They also describe experiments to determine the genomic basis of karyotype variation in this organism, and identify a candidate genomic region that encompasses the long sought after target for the agrochemical Metalaxyl. In a similar way that the original genome publication revolutionized how researchers worked with P. infestans, this chromosome level genome assembly has promise to be of similar utility to the community. The work is of excellent quality, the manuscript is generally well written, and it will have high impact. There are only minor issues for the authors to address.

Reviewer #4: This script describes, in great detail, a major advance in Phytophthora genomics and biology, offering the first virtually complete telomere to telomere genome assemblies of P. infestans that will continue to advance studies of this important pathogen for years to come. Furthermore, the study examines several key aspects of the genome including; stability & distribution of genes, repetitive sequences, karyotype vs phenotype, copy number and expression level polymorphisms (ELPs) and mapping and control of specific trait (metalaxyl resistance). The study is robust and authors have supported hypotheses, where required, with validation by independent qPCR experiments.

The data are excellent and the analysis full and appropriate. There are a couple of areas that I felt could be explored:

1. A Peronspora full-length T2T assembly was published on BioRxiv in 2021 with indications of strong synteny and genomic organisation to Phytophthora genomes (Fletcher et al 2022). Although a comparison of the assemblies is not required at this stage of the current study, I feel the work of Fletcher et al should be acknowledged and the value of downy mildew versus Phytophthora comparisons flagged as valuable for future studies. This could, for example, be built into the discussion around L671-8.

Fletcher K, Shin OH, Clark KJ, Feng C, Putman AI, Correll JC, Klosterman SJ, Van Deynze A, Michelmore RW, 2022. Ancestral Chromosomes for Family Peronosporaceae Inferred from a Telomere-to-Telomere Genome Assembly of Peronospora effusa. Mol Plant Microbe Interact 35, 450-63.

2. Much of the discussion presumes that the behaviour of the laboratory strains and their progeny is typical of a population of diploid strains in nature. How can authors be sure that this is the case? Caution needed as these strains have been maintained on agar under in vitro conditions for decades. It could be argued that this either represents stressed conditions (away from natural habitat) or ‘relaxed conditions’ with no plant infection or environmental stressors. I feel some acknowledgement of this is required. This in no way devalues the excellent data but a proposal to test these features of genome biology in field populations could be added.

It would have been nice to include studies on that could advance our understanding of the biology and genetics of mating type amongst the progeny but perhaps that is planned for a future study?

L372-387 shows that the tested phenotypes of F1 isolates were remarkably robust to changes in karyotype. It would have been interesting to test more traits such as virulence but again that was beyond the scope of an already comprehensive paper.

Will the genome assembly be available somewhere that can be accessed by a Genome Data Viewer for the community to exploit? It would have been nice to have compared strain 1306 to T30-4 but I realise that this was probably beyond the remit of this paper.

Part II – Major Issues: Key Experiments Required for Acceptance

Reviewer #1: (No Response)

Reviewer #2: (No Response)

Reviewer #3: (No Response)

Reviewer #4: None

Part III – Minor Issues: Editorial and Data Presentation Modifications

Reviewer #1: 57: I’m not convinced that this a truly “novel” observation. The authors themselves cite a previous study detailing this observation (lines 79-80)

112: please specify what type of variation you are referring to here

149: where did the genetic data for the 3 alternate parental strains come from? I see in the methods that they were sequenced using Illumina short reads and mapped to the reference, perhaps a brief explanation of that procedure is warranted here

152: where did the data for these 82 progeny come from?

209-210: the syntax of this sentence is non standard: it should read “549 and 128 genes were predicted to encode .. etc”

213: the term “gene cluster” is more commonly used to describe metabolic gene clusters where genes encode different reactions in the same metabolic pathway. To avoid confusion, please describe these genes using a different term or with additional descriptors, perhaps “densely arranged clusters of genes”. Also, it would be useful to provide a definition of how these clusters of genes were defined both at its first mention and in the methods.

Figure 1: should the X axis be labeled ‘alternate allele’ and not ‘alternate read’, as it is in some later figures?

220-221: what data support this assertion?

244-250: the observation that P. infestans has 7.6 times the amount of tRNAs compared with P. ramorum is very interesting, and the assembly generated by the authors provides a great opportunity to study the dynamics of tRNA evolution in this pathogen. How genetically variable are different copies from the same tRNA family, and are there some families that are more variable than others?

378-381: The authors observe that diploids grow faster than any other karyotype. However, there appear to be pretty large differences in sample sizes between the different karyotype categories (e.g., there are ~40 diploids but only two 4N isolates). How might differences in sample sizes have impacted your ability to detect differences between groups? And can you please indicate the sample size of each Karyotype category in Figure 6?

392: please include some details about how many strains were examined for RNA expression exactly.

467-468: did these events actually result in changes to transcript abundance? It would be useful to report the fold changes of these examples specifically.

480: I’m a little confused about which ‘silent’ genes you are referring to here: perhaps some context would be helpful. Also, to which ‘family’ are you referring? Is it the same family or just any family at all? Please clarify.

681: do you mean the ease at which these mutations arose, or do you really mean to refer to the ease at which they were detected?

902-903: can you please provide some brief details about how this script works exactly?

Reviewer #2: (No Response)

Reviewer #3: I suggest that the authors change ‘chemical resistance’ to ‘chemical insensitivity’ in the title. This will be more in line with terminology in the rest of the manuscript.

Line 341 reports that Fig. 5C shows F1 strain 20.23, an aneuploid diploid (2N1-). In a closer examination of this figure panel, it looks as if it should be 2N3- (chromosomes 7, 8, and 15). Can the authors please check this.

Line 397. The authors state that it is ‘more common for mRNA levels to be higher in polyploids’. The normalized data (not just the fold-change) supporting this statement need to be shown in a supplementary table. The same comment applies to RNAseq comparisons between strain 618 and 1306.

Line 655. The authors speculate that there may be a selective advantage to nuclei undergoing ploidy reduction in culture or within a lesion. However, this is not borne out in reality, as the authors note in Introduction lines 96-97 that the majority of field isolates are triploid. It may be best to remove this speculative statement on lines 654-655.

Line 695-696. Some more detail on the isolates used in this study is needed. What host plants were they originally isolated from, what year, and what culture collection(s) are they held in?

Line 708. It is surprising that the disease score assays were only carried out on potato tuber slices. Late blight is primarily a foliar disease, so why was leaf infection not used? Sliced tuber tissue is effectively wounded tissue, so removes the early infection events from the analysis of pathogenicity. Tuber slice assays will only reveal the extent to which hyphae can colonize tissue. This should be pointed out in the Results.

Line 754. Literature citation for the ‘Chicago method’.

Line 755. Geographical location for Dovetail.

Line 756. Geographical location for Cofactor Genomics.

Line 763. Insert ‘.’ into Chimericognizer [90] The data

Line 765. Literature citation for BUSCO.

Line783. Markers that showed

Line 784. Literature citation for genotype-corrector.

Line 858-859. Which Illumina sequencer was used? HiSeq, NextSeq, MiSeq?

Line 880-881. More detail is needed for the RNA extraction method than just ‘using phenol/chloroform’. Please describe the method or give a literature citation for it. The same comment applies for the polyacrylamide-urea gel electrophoresis; please describe the conditions or provide a literature citation.

Line 912. Manufacturer needed for DNase.

Line 914. Is the full gene ID PITG_09862? What is the function of the protein encoded by this gene? Can a GenBank accession be given for it?

The formatting of references should be checked. Some organism names need to be in italic font.

The figures in the manuscript are in some instances necessarily complex. As a general comment there needs to be more detail given in the figure legends to help the reader interpret them without having to delve into the article text to find the information.

Figure 1. What does ‘density’ on the Y axis describe? What is the red trace line in each graph? What does the pink shaded area represent?

Figure 2F. What do the orange or red horizontal lines in some of the chromosome bars represent?

Figure 3. The legend for this figure is particularly minimal. There needs to be more detail about what is shown in the layout for each chromosome. State the significance of the ‘forked’ mapping seen for some linkage groups.

Figure 5D legend. Line 1285. Correct (Chr3, Chr4, Chr7, Chr9). Line 1287 states that this is for Chr3, but Chr9 is shown in the figure.

Reviewer #4: There are a few minor points to address as listed below.

Are all gene names in this study the same as those from other work with PITG gene names? I see this in mentioned in L867 but it is not explicit. Could this be clarified and if not the case, then a look-up table needs to be included to allow comparisons.

Fig S5 and text L292-300. This figure is a great idea but the key SSR markers in the 12-plex set of Li et al (2013) including some from Lees et al (D13, Pi4B and Pi63) are missing. It would be helpful to also include the SSRs below. The claim in L298 that “while eight chromosomes lacked any marker” may be untrue if these additional 8 markers are added. Similarly, with L630-1 “only sample selected chromosomes”. Lastly, an explanation of the letters following the RG57 marks in Fig S5 would be helpful and please state what the grey bars on the chromosomes represent.

SSR11Fwd-TTAAGCCACGACATGAGCTG

SSR11Rev-GTTTAGACAATTGTTTTGTGGTCGC

D13Fwd-TGCCCCCTGCTCACTC

D13Rev-GCTCGAATTCATTTTACAGACTTG

SSR8Fwd-AATCTGATCGCAACTGAGGG

SSR8Rev-GTTTACAAGATACACACGTCGCTCC

SSR4Fwd-TCTTGTTCGAGTATGCGACG

SSR4Rev-GTTTCACTTCGGGAGAAAGGCTTC

SSR6Fwd-GTTTTGGTGGGGCTGAAGTTTT

SSR6Rev-TCGCCACAAGATTTATTCCG

SSR2Fwd-CGACTTCTACATCAACCGGC

SSR2Rev-GTTTGCTTGGACTGCGTCTTTAGC

Pi4BFwd-AAAATAAAGCCTTTGGTTCA

Pi4BRev-GCAAGCGAGGTTTGTAGATT

Pi63Fwd-ATGACGAAGATGAAAGTGAGG

Pi63Rev-CGTATTTTCCTGTTTATCTAACACC

Li Y, Cooke DE, Jacobsen E, Van Der Lee T, 2013. Efficient multiplex simple sequence repeat genotyping of the oomycete plant pathogen Phytophthora infestans. Journal of Microbiological Methods 92, 316-22.

L456 “no polymorphisms were sometimes detected” cannot detect something that does not exist - suggest “a lack of polymorphism in some case which may implicate…”

L695-6 year of isolation of the strains should be added.

L945 dna vs DNA

L1122 and L1000 italics missing

PLOS authors have the option to publish the peer review history of their article (what does this mean?). If published, this will include your full peer review and any attached files.

Do you want your identity to be public for this peer review? For information about this choice, including consent withdrawal, please see our Privacy Policy.

Reviewer #1: Yes: Emile Gluck-Thaler

Reviewer #2: No

Reviewer #3: No

Reviewer #4: No

Figure Files:

Data Requirements:

Reproducibility:

References:

---

## [Editor Report · Decision Letter 1]

9 Sep 2022

Dear Dr. Judelson,

We are pleased to inform you that your manuscript 'Karyotype variation, spontaneous genome rearrangements affecting chemical insensitivity, and expression level polymorphisms in the plant pathogen Phytophthora infestans revealed using its first chromosome-scale assembly' has been provisionally accepted for publication in PLOS Pathogens.

Before your manuscript can be formally accepted you will need to have the genome assembly as well as annotation deposited at NCBI (or other public databases) and to complete some formatting changes, which you will receive in a follow up email. A member of our team will be in touch with a set of requests.

Best regards,

Michael F. Seidl

Guest Editor

PLOS Pathogens

Bart Thomma

Section Editor

PLOS Pathogens

Kasturi Haldar

Editor-in-Chief

PLOS Pathogens

orcid.org/0000-0001-5065-158X

Michael Malim

Editor-in-Chief

PLOS Pathogens

orcid.org/0000-0002-7699-2064

---

## [Editor Report · Acceptance letter]

5 Oct 2022

Dear Dr. Judelson,

We are delighted to inform you that your manuscript, "Karyotype variation, spontaneous genome rearrangements affecting chemical insensitivity, and expression level polymorphisms in the plant pathogen Phytophthora infestans revealed using its first chromosome-scale assembly," has been formally accepted for publication in PLOS Pathogens.

Best regards,

Kasturi Haldar

Editor-in-Chief

PLOS Pathogens

orcid.org/0000-0001-5065-158X

Michael Malim

Editor-in-Chief

PLOS Pathogens

orcid.org/0000-0002-7699-2064